# Reaction-conditioned generative model for catalyst design and optimization with CatDRX
Apakorn Kengkanna [1], Yuta Kikuchi [1], Takashi Niwa[2] & Masahito Ohue [1] ✉

Designing effective catalysts is a key process for optimizing catalytic reactions to reduce time and waste during scale-up. Recently proposed approaches, including generative models, show promise in identifying new catalysts. However, they are mostly developed for specific reaction classes and predefined fragment categories without considering reaction components, limiting the exploration of novel catalysts across reaction space. Here, we present CatDRX, a catalyst discovery framework powered by a reaction-conditioned variational autoencoder generative model for generating catalysts and predicting their catalytic performance. The model is pre-trained on a broad reaction database and fine-tuned for downstream reactions. Our approach achieves competitive performance in both yield and related catalytic activity prediction. Additionally, it enables effective generation of potential catalysts given reaction conditions by integrating optimization toward desired properties and validation based on reaction mechanisms and chemical knowledge, as demonstrated in various case studies. This work helps facilitate and advance catalyst design and discovery for chemical and pharmaceutical industries.

Designing and developing catalysts is a crucial process for increasing the efficiency of industrial chemical processes, minimizing waste, and enhancing a sustainable society[1–4]. Typically, catalyst development is considered a multi-step process that can take several years, from initial screening to industrial application[5,6]. To minimize resources and efforts associated with scale-up, primary screening is a critical step for identifying promising catalyst candidates[7].

In primary screening, researchers conduct extensive experiments across reaction conditions to evaluate catalyst or ligand performance[8]. While some experiments may involve simple screening with limited catalyst sets, the real-world scenario is often more complex, covering large combinatorial sets of reaction conditions and additional optimization steps, requiring tremendous effort to navigate sophisticated chemical space[9,10]. Furthermore, evaluating catalyst effectiveness involves different efficiency metrics depending on the type and nature of catalysis[11]. Beyond screening existing catalyst libraries, the design of novel catalyst candidates is also essential for advancing new chemical reactions and synthetic methodologies, particularly in this current era where green catalysis is gaining increasing attention[12–14].

On account of that, conducting catalyst design using conventional experimental methods by trial-and-error is costly and time-consuming[15]. Computational chemistry calculations, such as density functional theory (DFT) and energy profile molecular dynamics, are then employed for catalysis study[16]. Although these methods demonstrate good results, they still require substantial computational resources and largely depend on empirical knowledge or theoretical assumptions[17,18]. Besides, their applicability is often limited, making them less transferable to different or large-scale systems[19].

With the advancement of artificial intelligence (AI) across various fields, AI has been introduced to drive the catalyst design and development[19–25]. Several machine learning (ML) techniques have been utilized for predicting catalytic performance for chemical reactions. Physics-based descriptors together with structural data and transition state profiles have been generally employed to represent catalysts for predictive model[26–30]. Some studies have employed deep learning models with less expensive molecular representations, including fingerprints[31–33] or molecular graphs[34]. Active learning has also been utilized to search the candidate catalyst space[35]. To alleviate the issue of limited datasets, recent studies introduce pre-trained models on large available reaction data for yield prediction by leveraging different input features, for example, string-based notations like SMILES[36,37], molecular graphs[38], or combined features[39]. Even though learning from a large amount of reaction data improves generalization, the screening process remains constrained by the availability of existing catalyst or ligand libraries, restricting the discovery of entirely new catalysts[40].

[1]Department of Computer Science, School of Computing, Institute of Science Tokyo, Kanagawa, Japan. [2]Graduate School of Pharmaceutical Sciences, Kyushu University, Fukuoka, Japan. ✉e-mail: ohue@comp.isct.ac.jp

To promote novel catalyst generation, genetic algorithms (GAs) approaches have been utilized for designing catalyst[41–46]. GAs offer optimization cycles based on natural selection to discover novel molecules with a target objective. Despite promising performance and validity of GAs, they are particularly suitable for systems with well-defined building blocks, and creative outputs are often limited by the finite size of the fragment database. Additionally, GAs require considerable evolutionary cycles and a reliable fitness function to effectively guide the optimization[41,47].

Recently, generative models have been proposed to advance catalyst development through inverse design strategies[41,48]. Numerous techniques have been studied for catalyst and transition metal complex generation, for instance, variational autoencoder model (VAE)[49–51], transformer-based language model[52], and diffusion generative model[53,54]. Despite their potential, those methods often overlook crucial reaction conditions, especially reactants, products, reagents, and reaction time, limiting their applicability across a broader range of reactions. Also, generating molecules outside training data and synthesizability remains an important issue. Although some efforts have been made to transform reaction conditions into a generative model for predicting suitable conditions[55], these models typically treat reaction conditions as fixed sets, thereby constraining the exploration of external conditions.

To address the limitations of prior generative models, which are often restricted to specific reaction classes, conditions, or a limited set of structural motifs, in this study, we introduce CatDRX, Catalyst Discovery framework based on a ReaXion-conditioned variational autoencoder (VAE) for catalyst generation and catalytic performance prediction. We summarize our contributions as follows:

- We propose CatDRX, a deep generative model for catalyst design and discovery under given reaction conditions, based on a joint VAE architecture by learning structural representations of catalysts and associated reaction components, aiming to capture their relationship on reaction outcomes. The model is pre-trained on a variety of reactions from Open Reaction Database (ORD)[56], facilitating broad reaction conditions, and then fine-tuned on downstream datasets.
- We conduct extensive experiments on multiple reaction classes to evaluate the model's performance in predicting both yield and related catalytic properties, enhancing the catalyst screening process. Additionally, we analyze its ability to generate catalysts using different sampling strategies, promoting broader exploration of the chemical space.
- We demonstrate inverse design in three case studies by integrating background knowledge filtering, optimization toward desired targets, and validation of catalyst candidates using computational chemistry, showcasing the model's practical utility in generating novel potential catalysts to accelerate the catalyst discovery pipeline.

## Results and discussion
### Method Overview
The overview of the study workflow is visualized in Fig. 1a. This study develops a generative and predictive model that learns structural representations of catalysts and associated reaction components in order to generate catalysts and predict their catalytic performance. The model is initially trained through a pre-training process on various reactions from Open Reaction Database (ORD), and then fine-tuned on downstream datasets using the whole pre-trained model including encoder, decoder, and predictor. The model is capable of designing novel catalysts as well as predicting reaction yields and related catalytic properties as an extension. With a unified training design, the model can be integrated with optimization techniques to generate optimized catalysts under given reaction conditions, enabling efficient exploration of the reaction space. The generated catalysts are validated using computational chemistry tools and background knowledge, demonstrating the effectiveness of the proposed approach.

Regarding the model architecture, the design is based on a jointly trained Conditional Variational Autoencoder (CVAE) integrated with property prediction using reaction components as conditions, as shown in

Fig. 1b. The model consists of three main modules, including the catalyst embedding module, the condition embedding module, and the autoencoder module. The catalyst embedding module embeds the catalyst matrix through a series of neural networks to construct a catalyst embedding. The condition embedding module learns other reaction components, which are reactants, reagents, products, and additional properties, such as reaction time, to form a condition embedding. These two embeddings are concatenated into a catalytic reaction embedding and passed to the autoencoder module, which includes encoder, decoder, and predictor. The encoder maps the input into a latent space of catalysts and chemical reactions. A latent vector is then sampled from this space and concatenated with the condition embedding to guide the decoder in reconstructing catalyst molecules with post-processing enhancement. The predictor uses the same latent vector and condition embedding to estimate catalytic performance. During pre-training, the predictor is trained to estimate percent yield and can be fine-tuned for downstream tasks. For other catalytic activities, an additional surrogate model can be introduced to predict other properties accordingly.

### Catalytic activity prediction performance
First of all, the predictive performance of the model is evaluated using downstream datasets. Fig. 2 reports the catalytic prediction performance in terms of root mean squared error (RMSE) and mean absolute error (MAE) in comparison with existing baselines. Additional performance metrics including the coefficient of determination ($R^2$) are summarized in Supplementary Note 5. Performance metrics for the comparative models are taken directly from their original publications when available. Otherwise, the models are reproduced for comparison.

Overall, the model demonstrates superior or competitive performance across various datasets, particularly in yield prediction, as the prediction module is directly incorporated during model pre-training. However, the model encounters challenges with certain datasets, especially those related to other catalytic activities. To investigate this, the chemical spaces of both reactions and catalysts are examined to assess domain applicability. For the reaction space, reaction fingerprints (RXNFPs)[57] with 256-bit embeddings are used to analyze the reaction samples. In parallel, the catalyst space is represented using 2048-bit ECFP4 fingerprints. Figure 3a, b presents the t-SNE embeddings of the reaction and catalyst chemical spaces. As observed, BH, SM, UM, and AH datasets show substantial overlap with the pre-training dataset, suggesting that the model benefits from transferred knowledge during the fine-tuning step. In contrast, datasets such as RU, L-SM, CC, and PS exhibit minimal overlap with the pre-training data, indicating different domains in reaction classes, resulting in reduced performance, especially for the CC dataset. Similarly, in the catalyst space, a large portion of CC catalysts are located outside the pre-training region, further reducing the effectiveness of transfer learning. Moreover, the CC dataset contains only a single reaction condition, which limits the model's ability to leverage condition-based knowledge and forces it to rely solely on the catalyst input. This results in an overly complex architecture and degraded performance. These observations suggest that expanding the diversity of reactions and catalyst databases would broaden the chemical space coverage, thereby providing a more comprehensive foundation to the model. Another aspect is about catalyst featurization. Currently, our model encodes catalysts using atom and bond types along with the adjacency matrix. However, features such as atomic charges and chirality may be important for accurately representing specific catalysts, such as asymmetric catalysis. In this study, for the AH dataset, according to the original sources[28,37], the catalysts are axially chiral but maintain an (S) chirality configuration and share the same axial configuration, making this aspect irrelevant here. For the PS dataset[46], all catalysts are represented in a string-based format with a chiral carbon but without explicit stereochemical configuration. The target property is enantioselectivity, measured by $\Delta\Delta G^{\ddagger}$, so the absolute product configuration is not directly relevant. Currently, the model does not include chirality information, so it can only focus on one stereoisomer of the outcome. Incorporating additional features could enrich representation and improve model learning. For example, encoding chirality configuration as

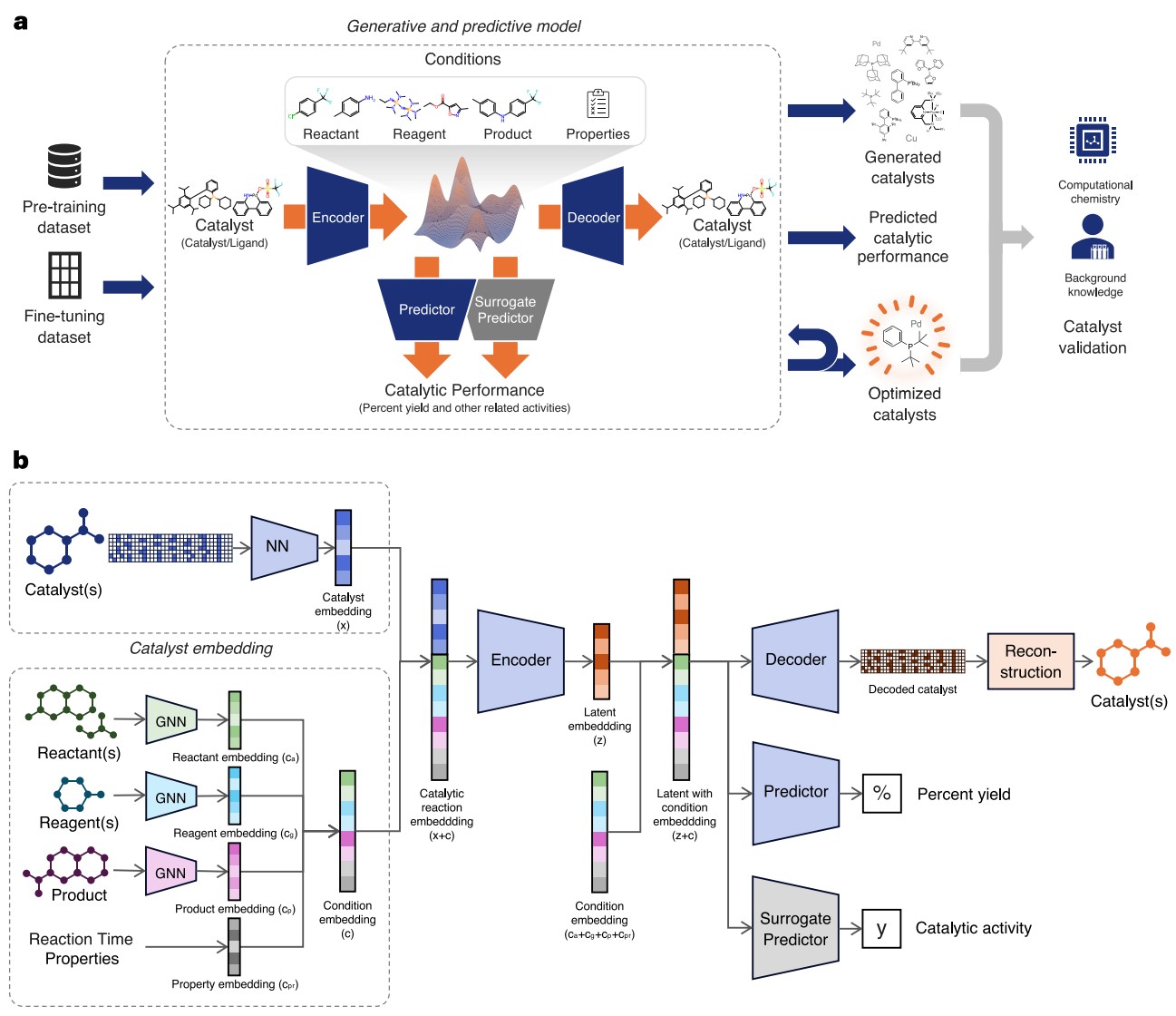

**Fig. 1 | Overview of CatDRX workflow and model architecture. a** CatDRX workflow. A generative and predictive model is introduced to learn information from catalysts and reaction conditions in order to generate promising catalysts and predict catalytic performance. The model is pre-trained on a pre-training dataset and further fine-tuned using downstream datasets. The model is integrated with an optimization and design module to generate promising catalysts with desirable properties. The results are validated through computational chemistry and background knowledge from chemists. **b** CatDRX model architecture. The model is designed based on a jointly trained Conditional Variational Autoencoder (CVAE) architecture, comprising catalyst embedding, condition embedding, and auto-encoder module with encoder, decoder, and predictor. Catalysts and reaction conditions are embedded separately, then combined as input to the encoder to learn latent representations. The decoder reconstructs catalysts from sampled latent vectors and condition embeddings, while the predictor estimates reaction yield. For other catalytic tasks, a surrogate predictor is fine-tuned accordingly.

part of the input conditions could enhance catalyst features, similar to the approach in ref. 34. Nevertheless, the model demonstrates promising performance in predictive tasks jointly learned alongside the generative task, presenting a viable approach for subsequent studies.

In addition to evaluating the complete model, ablation studies are conducted to support the importance of all proposed modules. Four alternative variants include a pre-trained model without augmentation, a pre-trained model with augmentation but without fine-tuning, a model without pre-training and augmentation, and a pre-trained model without fine-tuning and augmentation. The results are summarized in Supplementary Note 5. According to the results, the completed CatDRX model outperformed other variants in general. Augmentation seems to have a small effect on model performance. Due to the variation of reaction classes with a limited number of reactions in the pre-training datasets, using only the pre-trained model on downstream tasks may result in domain incompatibility, making the fine-tuning process necessary at this

moment. However, pre-training helps the model learn well, particularly for difficult dataset splittings. Additional results comparing the model fine-tuned from a pre-trained model and the model trained only on the downstream task are shown in Supplementary Fig. S6. The models fine-tuned from a pre-trained model achieved better RMSE performance on many datasets, indicating that they benefit from the knowledge learned during the pre-training stage. On the other hand, during fine-tuning on the downstream dataset, signs of forgetting pre-trained knowledge emerged. We monitored performance on both the downstream test set and a held-out pre-training test set across epochs as shown in Supplementary Fig. S7 and Table S7. While the model performed well on the downstream dataset, its performance degraded on the pre-training test set, indicating forgetting. This occurs because fine-tuning prioritizes downstream performance. Future work could explore strategies that balance performance to mitigate forgetting. However, all of these results confirm the performance of our model with the full options.

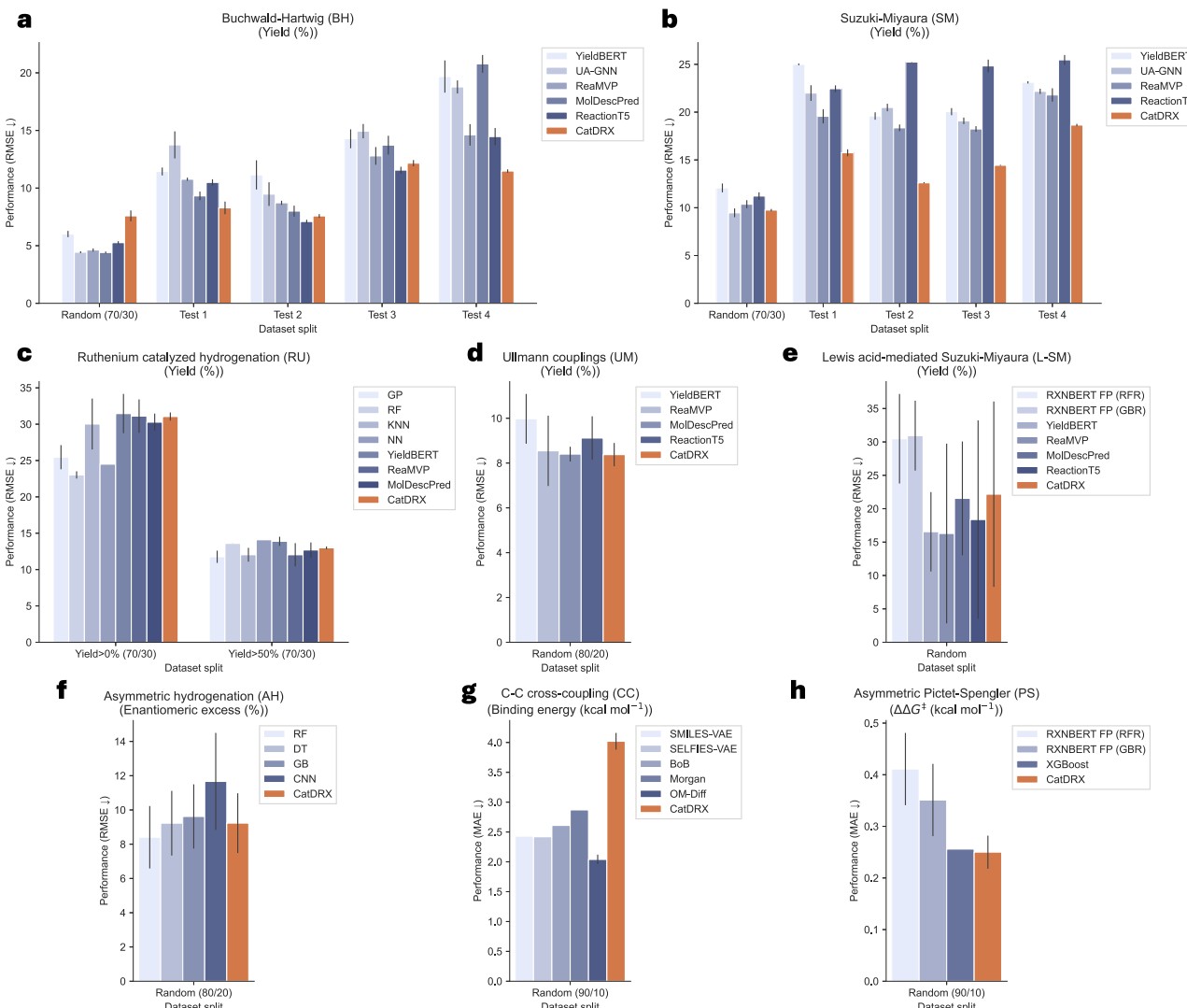

**Fig. 2 | Model prediction performance on various downstream datasets.** Metrics are reported in RMSE or MAE according to dataset. The orange color represents our model performance. The standard error lines are indicated where available. **a** Buchwald-Hartwig (BH) dataset[29,39]. Baselines: YieldBERT[36], UA-GNN[88], ReaMVP[39], MolDescPred[38], and ReactionT5[71]. The Test 1-4 data splits follow the additive-based out-of-sample split[29]. **b** Suzuki-Miyaura (SM) dataset[39,89]. Baselines: YieldBERT, UA-GNN, ReaMVP, and ReactionT5. The Test 1-4 data splits follow the ligand-based out-of-sample split[39]. **c** Ruthenium catalyzed hydrogenation (RU) dataset[90]. Baselines: GP (Gaussian processes), RF (Random forest), KNN (K nearest neighbors), NN (Neural network)[90], YieldBERT, ReaMVP, and MolDescPred.

**d** Ullmann couplings (UM) dataset[91]. Baselines: YieldBERT, ReaMVP, MolDescPred, and ReactionT5. **e** Lewis acid-mediated Suzuki-Miyaura (L-SM) dataset[8]. Baselines: RXNBERT FP[57] (RFR (Random forest regressor), GBR (Gradient boosting regressor)), YieldBERT, ReaMVP, MolDescPred, and ReactionT5. **f** Asymmetric hydrogenation (AH) dataset[28]. Baselines: RF (Random forest), DT (decision tree), GB (extreme gradient boosting), and CNN (Convolutional neural networks)[28]. **g** C-C cross-coupling (CC) dataset[59]. Baselines: SMILES-VAE, SELFIES-VAE, Morgan fingerprint[49], BoB (Bag of bonds)[92], and OM-Diff[53]. **h** Asymmetric Pictet-Spengler (PS) dataset[46]. Baselines: RXNBERT FP (RFR, GBR), and XGBoost[46].

## Latent space analysis

To observe how the model learns the distribution of reactions, latent spaces are investigated. After fine-tuning, the latent space is extracted and visualized using t-SNE technique. Suzuki-Miyaura (SM)–Test 1 dataset is taken as an example for this study, and the results of reaction dataset embedding are visualized in Fig. 4. We can observe that the model effectively learns the structural information and clearly constructs distinct latent spaces for each ligand type (Fig. 4a). Additionally, it organizes the latent space according to true percent yield, as indicated by a noticeable gradient trend (Fig. 4b). Furthermore, by incorporating reaction information, the latent space regions corresponding to different ligands are well constructed into small clusters according to the respective class of components, as shown in the Supplementary Fig. S8.

An additional analysis is conducted by randomly sampling the latent space under specific conditions. Ten distinct conditions are selected from

the same dataset, and for each condition, one thousand catalysts are randomly generated following the generation process and embedded again with the corresponding condition to obtain embedding vectors for visualization. The t-SNE plots visualizing this embedding space are shown in Supplementary Fig. S9. The latent vectors effectively embed condition-specific features, and apparent yield trends can be observed within each condition. These results demonstrate that the model appropriately captures reaction information and constructs a structured chemical latent space, thereby supporting exploration for optimal catalysts with reaction conditions.

## Catalyst generation performance

Catalyst generation performance is evaluated using different sampling schema to assess the generative capability. Four sampling strategies are compared, including random latent vector and random condition from the training set (**Random latent + Random condition**), latent vector sampled

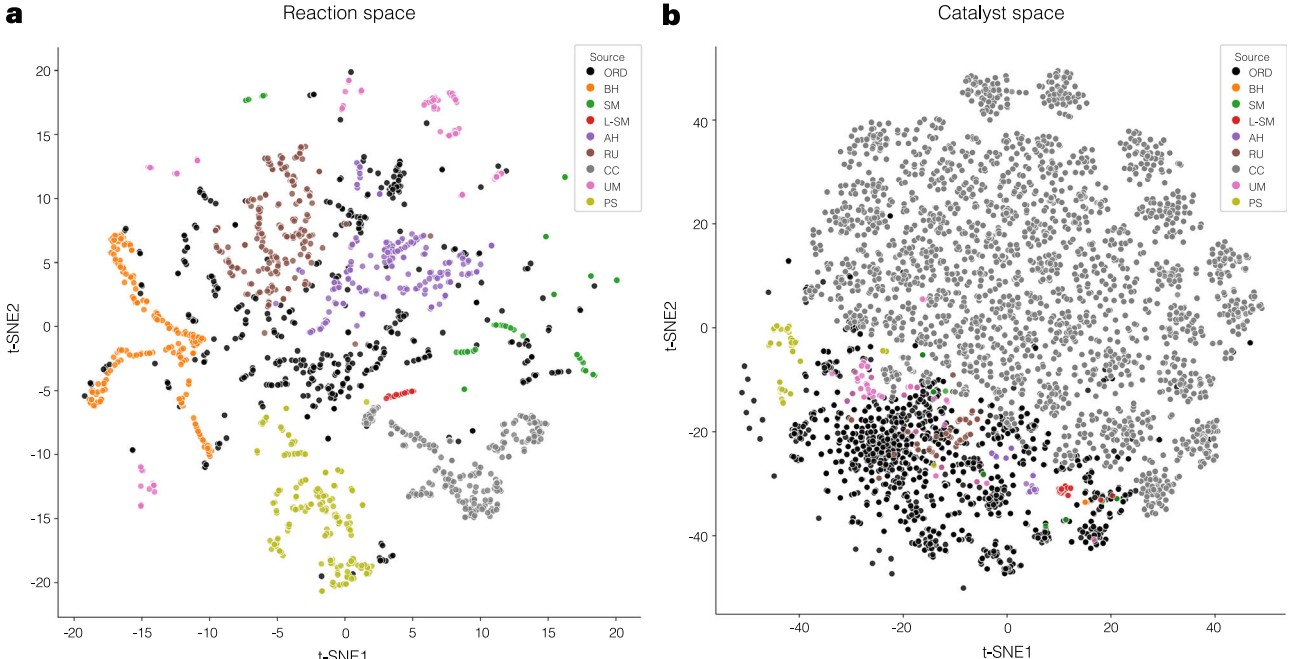

**Fig. 3 | Embedding of reaction and catalyst chemical space. a** Embedding of reaction space. 500 reactions randomly sampled from ORD and at most 300 reactions from each dataset are embedded using the RXNBERTFingerprintGenerator (256-bit) from reaction SMILES[57]. The t-SNE is performed to visualize the reaction space, labeled by the data source. **b** Embedding of catalyst space. All unique catalysts are retrieved from reaction data and encoded using ECFP4 fingerprint (2048-bit). The t-SNE embedding is performed to visualize the catalyst space, labeled by the data source.

around an existing catalyst and random condition from the training set (**Around sample + Random condition**), and latent vector sampled around an existing catalyst and its corresponding condition (**Around sample + Sample condition**) with and without the post-processing step (**w/o post-processing**). The post-processing step involves modification and correction techniques applied during molecule reconstruction using chemical rules to improve validity, as explained in detail in Model Architecture section. Ten thousand molecules are generated for each sampling scheme, and evaluation metrics are computed and visualized, as shown in Fig. 5 and Supplementary Table S8. The definition of task-specific validity (Valid (Task)) varies depending on the specific dataset. For L-SM and PS dataset, a catalyst is valid only if it is a single molecule. For SM (Random) dataset, validity requires the presence of the Pd(OAc)$_2$ complex. For CC dataset, a valid catalyst must contain three fragments, including a single metal atom coordinated with two ligands.

Based on the results, when considering latent space sampling, in terms of general validity, the majority of generated molecules are valid, primarily due to the incorporation of the post-processing step. This step is necessary and widely adopted in atom-based generative models, as omitting it can significantly reduce the number of valid molecules. When evaluating task-specific validity, a slight drop is observed for the **Around sample** scheme, while a more large decline is evidently appeared for the **Random latent** scheme. This can be attributed to the diversity of catalyst structures in the pre-training dataset, including both single and multiple-fragment molecules. Regarding uniqueness and novelty, these two metrics exhibit similar trends and both reduce as a consequence of low validity. For diversity, although the trend is not clearly observed across all schemas, the generation process still produces high diversity, which is desirable for a generative model. The **Random latent** scheme tends to generate a wide variety of catalyst structures with high uniqueness and novelty, but most of them may not conform to the task-specific validity criteria, as reflected in the lower similarity to a nearest neighbor in training set. In contrast, the **Around sample** scheme is more reliable, producing molecules that are valid and close to the training data. Although this approach may bring down

uniqueness, novelty, and diversity in some ways, task-specific validity remains within an acceptable range.

The effect of conditions is particularly evident in the PS dataset, which contains multiple conditions. Sampling with **Around sample + Sample condition** scheme tends to reduce uniqueness and diversity, likely due to the narrower generation scope, but gives little high task-specific validity compared to random condition. On the other hand, even when **Random condition** is used, the model still maintains task-specific validity and similarity to the training set. This suggests that the model has effectively learned a broader representation of condition space.

Lastly, similar to the model prediction performance, the ablation study is extended to model generation performance by comparing a model fine-tuned on a pre-trained model with a model trained only on the downstream dataset using PS dataset. The results are summarized in Supplementary Fig. S10. Both models achieved high validity with post-processing support. The fine-tuned model produced more unique and novel catalysts, particularly near known structures, likely due to broader knowledge from pre-training. The downstream-only model showed slightly higher task-specific validity, reflecting its narrower scope. Regarding diversity-related metrics, both models show similar trends. In summary, while the downstream-trained model attains good validity in generated catalysts, it exhibits lower uniqueness and novelty compared to the fine-tuned model.

### Case studies
We present three case studies to demonstrate the usability of CatDRX in designing and optimizing novel catalysts. The results are further validated by background knowledge and computational chemistry tools to support the model's potential.

**Lewis acid-mediated Suzuki-Miyaura cross-coupling (L-SM).** The first case study is the Lewis acid-mediated Suzuki-Miyaura cross-coupling reaction dataset from ref. 8. The researchers conducted ligand screening and optimization for a palladium catalyst using percent yield determined by $^{19}$F NMR yield as a measurement, and the reaction scheme

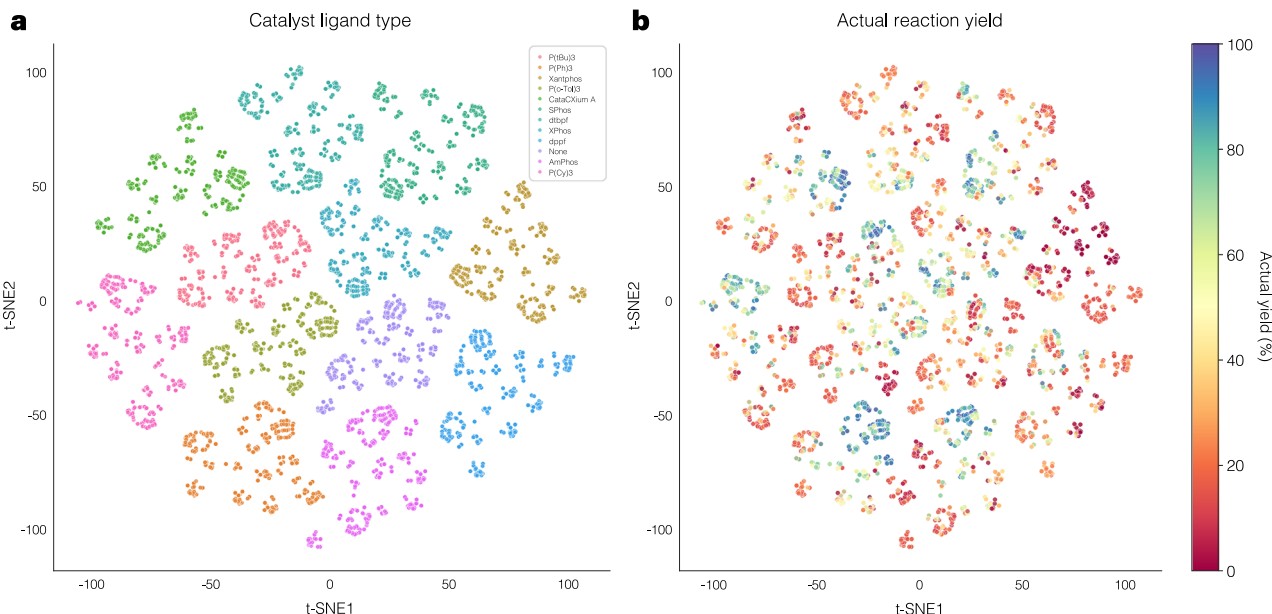

**Fig. 4 | Latent space embedding of the Suzuki-Miyaura (SM)–Test 1 reaction dataset with respect to catalysts and true yield.** The embedding visualized using t-SNE represents all reactions in the Suzuki-Miyaura (SM)—Test 1. **a** The t-SNE map of latent space embedding color-coded by catalyst ligand type. **b** The t-SNE map of latent space embedding color-coded by actual percent yield.

is shown in Fig. 6a. In this task, the goal is to design potential ligands for this catalytic process that can achieve high product yields.

Initially, we conduct experiments using various models, including simple fingerprint-based models, deep learning models, and our generative model with additional variations utilizing surrogate random forest and gradient boosting regressors. The results are shown in Supplementary Note 9. Although some deep learning models achieve the great results on the test set, they struggle to perform consistently on the training set as displayed in Supplementary Fig. S11–S12. In contrast, our model with a random forest regressor demonstrates competitive performance across both sets in both RMSE and $R^2$, leading to selecting it as our representative model. We decide to use the average predicted results from three models as the final prediction, and the scatter plot showing the model performance for the entire dataset is displayed in Fig. 6b.

For ligand generation, two approaches are tested, including generation at random constrained by the training space and generation around training molecules to emphasize structural similarity. Based on the background in catalysis and ligand availability, specific filtering principles are considered for identifying suitable ligands for this reaction. Molecules should contain nitrogen or phosphine atoms, as these elements play a key role in coordinating with metal complexes. Phosphines capable of binding to metals should possess three covalent bonds. Phosphines bearing a P-H bond are generally too reactive to function as ligands, often causing side reactions and degradation. Additionally, phosphines within three-membered rings are synthetically challenging to produce. These filtering criteria not only help increase the number of viable ligands but also promote synthetic accessibility. The example filtered generated ligands are displayed in Fig. 6d with the filtering rules and statistical results in Supplementary Table S9–S10. After filtering, generation around training molecules produces ligands with greater similarity to known structures, resulting in slightly lower diversity compared to random generation.

By the way, these generation methods do not consistently produce ligands with high yields. To emphasize high-yield ligands, we employ an optimization-based approach for generating them. Similar to generation, constraint rules are applied during optimization as penalties when violated. We also incorporate similarity measures relative to the training dataset to balance diversity and relevance. The example filtered optimized ligands are displayed in Fig. 6d. To compare with previous methods, we plot the

distribution of predicted yields as shown in Fig. 6c. Ligands generated using the optimization show improved validity and clearly a higher portion of predicted high yields compared to random generation. Nonetheless, a limitation remains as the optimized ligands do not achieve the highest yield seen in the dataset, possibly due to the limited accuracy and size of the dataset. Even with limited predictive accuracy, models can still provide useful trends for analysis, as suggested by a previous study[58]. Therefore, we further investigate ligands predicted as high yield to support the results. For generations around training molecules, most of the generated ligands resemble the structures of di(*tert*-butyl)phenylphosphine and amphos, which show high yield from the experiment. For generation at random and optimization, although diethylphenylphosphine performed poorly in the experiment, the model identifies dipropylphenylphosphine as a high-yield candidate, which is an interesting outcome for further experimental validation.

Lastly, we perform an optimization starting from an existing ligand to demonstrate how the model learns to explore the latent space for ligand improvement. Ligands with low experimental yields are selected, and the search is conducted around their embeddings. The results are presented in Fig. 6e. This optimization attempts to identify ligands with higher predicted yields within the local latent space. Some optimized ligands display high structural similarity to the starting ligand, suggesting that these variants could serve as practical candidates for guided modifications to enhance reaction yields.

**C-C cross-coupling (CC).** Another case study is the C-C cross-coupling reaction. This dataset includes various ligand combinations with six different transition metals for a family of C-C cross-coupling reactions, represented by an $L_1$-$M$-$L_2$ structural motif, where $L$ denotes a ligand and $M$ denotes a transition metal. The diagram of this catalytic cycle is illustrated in Fig. 7a. According to previous studies[59], the goal is to identify potential catalysts relevant to Suzuki-Miyaura cross-coupling reactions that can bind to substrates with an appropriate binding energy range of $-32.1$ to $-23.0$ kcal mol$^{-1}$. Binding energies are analyzed using a viable catalytic cycle descriptor via a volcano plot and calculated through a DFT approach[59]. In this case, an additional surrogate predictor is introduced to learn in the latent space during fine-tuning to predict the calculated binding energy. Based on the model's prediction performance and latent space analysis, the model may not achieve optimal results here, since the

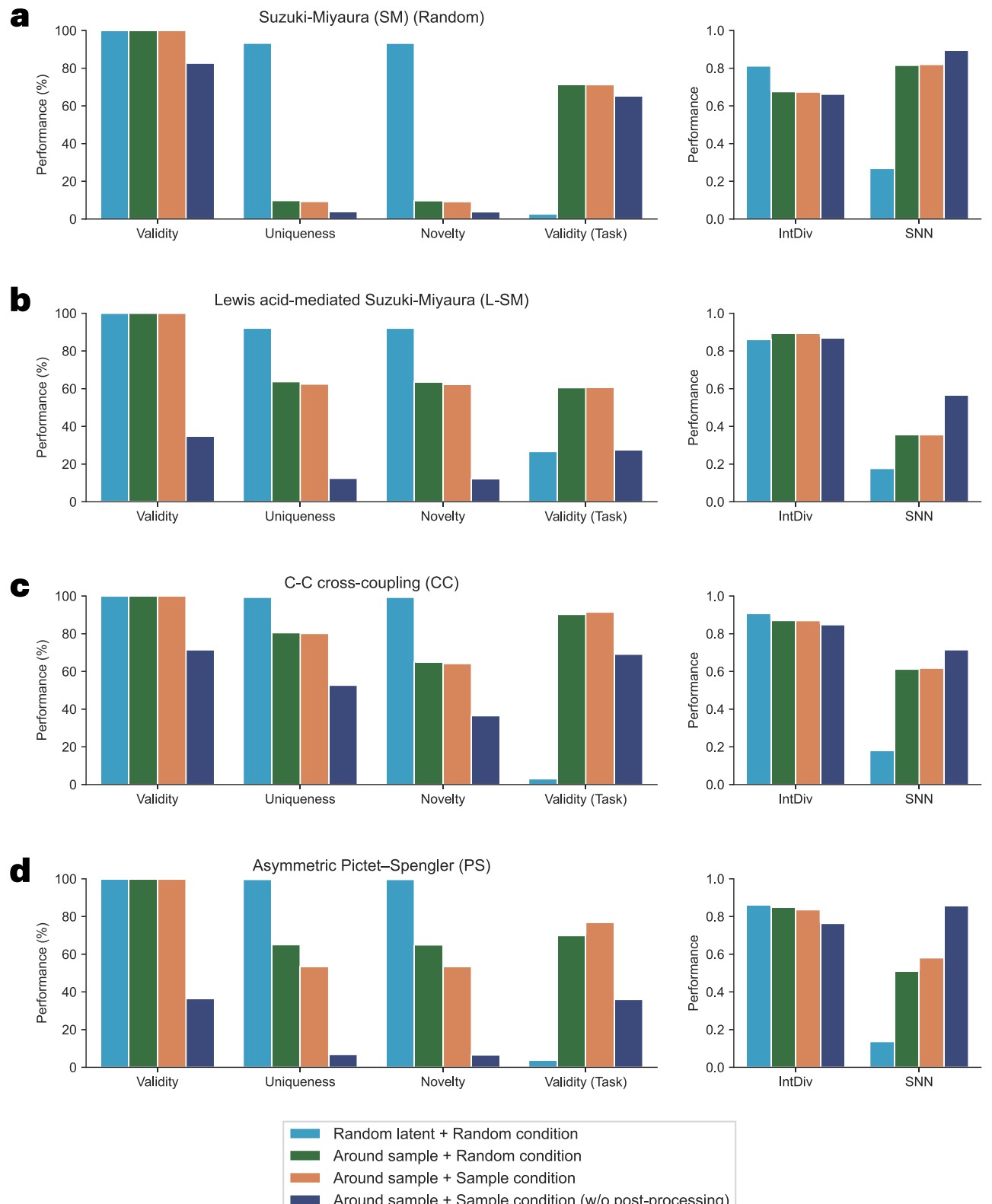

**Fig. 5 | Model generation performance using different sampling schema on four datasets.** Validity (Task) is measured differently based on the specific criteria defined for each dataset. IntDiv means internal diversity. SNN means similarity to a nearest neighbor. **a** Suzuki-Miyaura (SM) (Random) dataset. **b** Lewis acid-mediated Suzuki-Miyaura (L-SM) dataset. **c** C-C cross-coupling (CC) dataset. **d** Asymmetric Pictet-Spengler (PS) dataset.

dataset includes only a single condition and thus does not benefit from yield prediction across diverse condition designs. In spite of that, the model is considered useful with prediction errors within an acceptable range, approximately 5 kcal mol$^{-1}$, following the previous work[53]. We

select the best model for binding energy prediction, and the predictive results for all reactions in the datasets are displayed in Fig. 7b.

After that, we employ an optimization process to generate candidate catalysts by setting the objective target around the mean of the desired

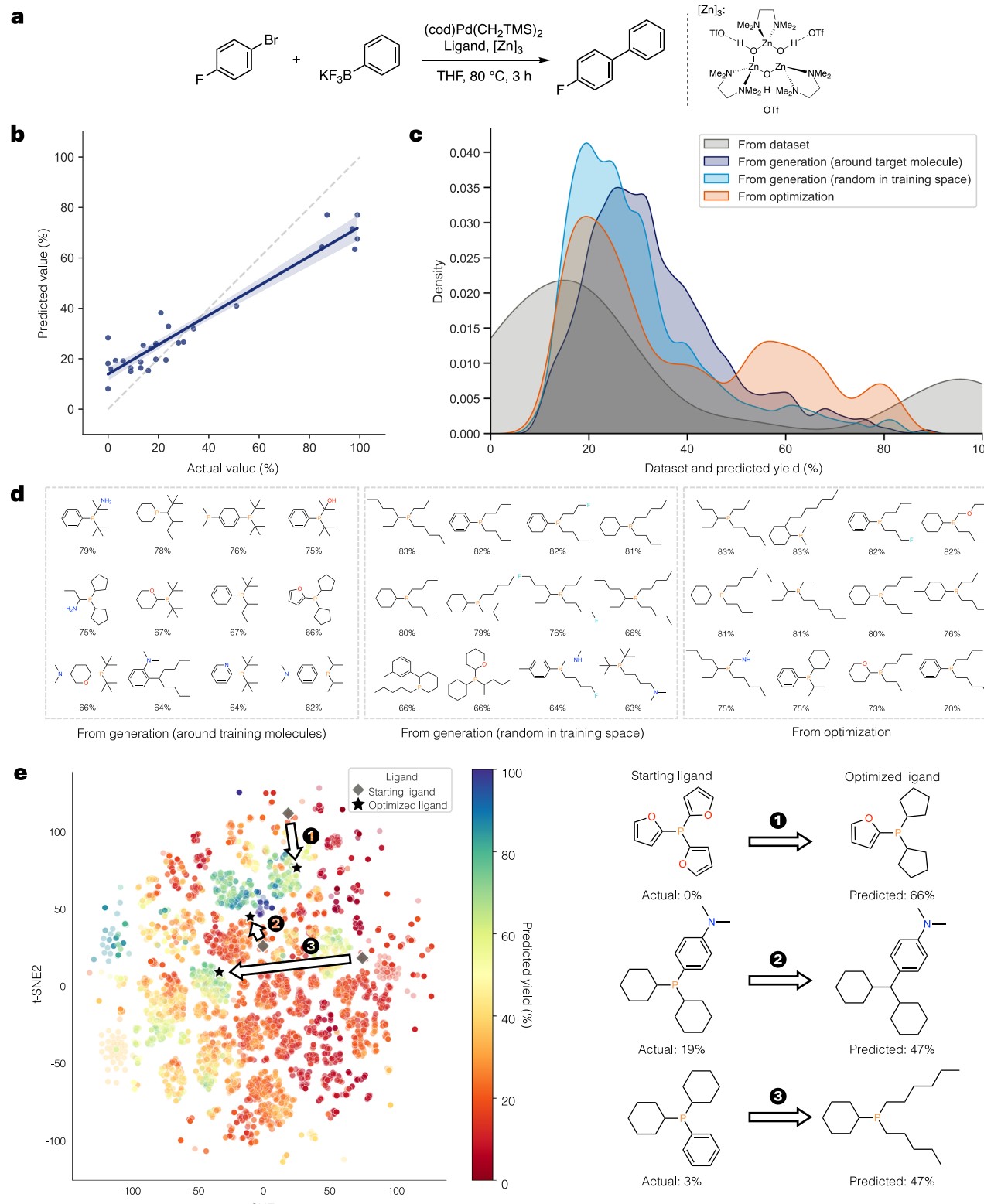

**Fig. 6 | Results on the case study of Lewis acid-mediated Suzuki-Miyaura cross-coupling (L-SM). a** Lewis acid-mediated Suzuki-Miyaura cross-coupling reaction used in optimization of ligand of palladium catalyst[8]. **b** Predictive performance of the model on the entire dataset. The blue line represents the regression line, and the light blue area represents the 95% confidence interval for that regression. **c** Distribution of the dataset and predicted percent yields from different generation approaches, visualized using kernel density estimation (KDE). **d** Examples of ligands generated by different approaches with their predicted percent yields. **e** Ligand generation using optimization. The starting ligand (gray diamond) and optimized ligand (black star) are mapped onto the latent space of randomly generated ligands, colored by predicted yield (left). The optimization transformation from the starting to optimized ligand is visualized with yield predictions and searching path indicated by numbers on the latent space (right).

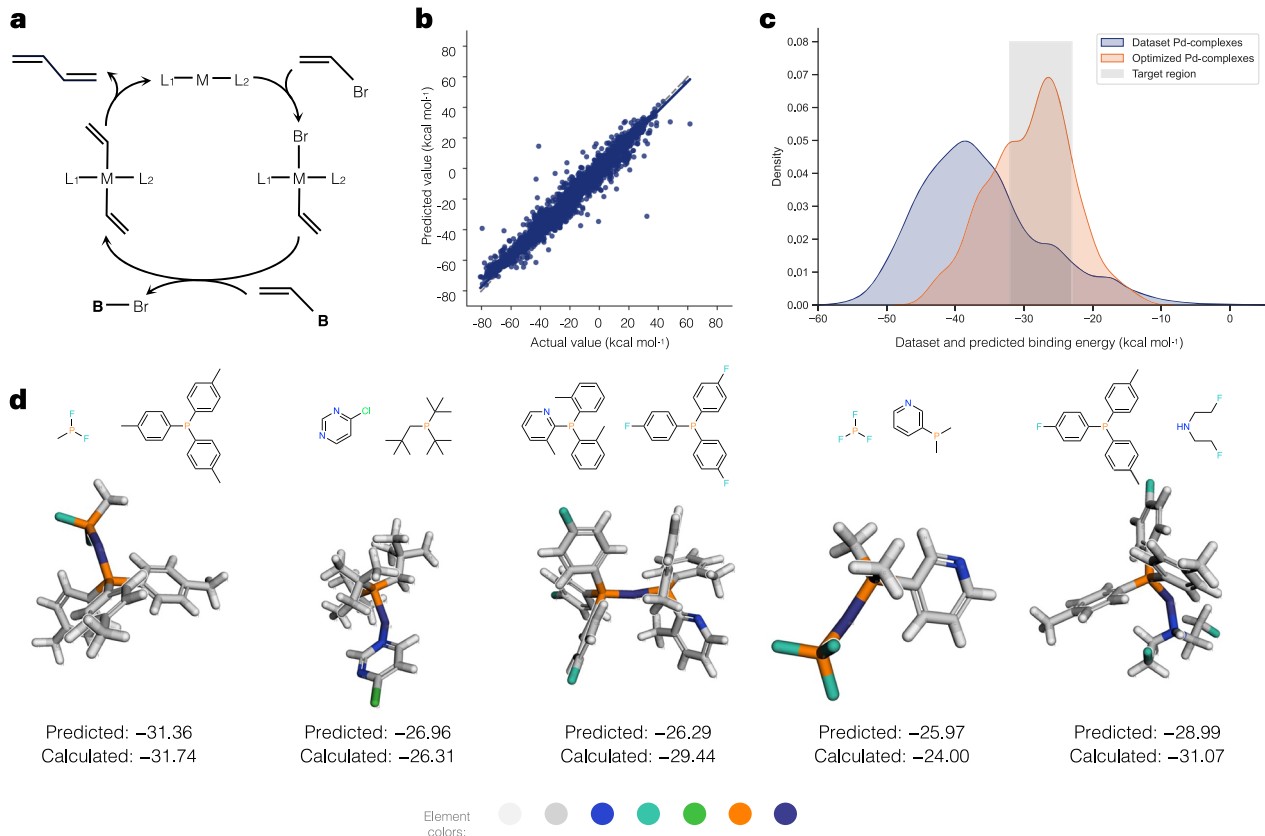

**Fig. 7 | Results on the case study of C–C cross-coupling reaction (CC). a** General catalytic cycle for C–C cross-coupling reactions. For the Suzuki-Miyaura reaction, **B** is [B(OH)$_2$(O$^t$Bu)]$^-$[59]. **b** Predictive performance of the model on the entire dataset. The blue line represents the regression line, and the light blue area represents the 95% confidence interval for that regression. **c** Distribution of the dataset and predicted binding energies from the optimization of Pd-complexes, visualized using kernel density estimation (KDE). The target binding energy range for effective Suzuki-Miyaura cross-coupling catalysts (−32.1 to −23.0 kcal mol$^{-1}$) is highlighted in gray. **d** Ligands and 3D structures of examples of novel generated Pd-complexes with predicted and validated binding energy (kcal mol$^{-1}$) with DFT.

binding energy range (−27.55 kcal mol$^{-1}$). To ensure the validity of the generated molecules, we perform optimization around target structures along with applying some penalties, including restriction in the number of fragments, absence of key atoms such as phosphorus, nitrogen, or oxygen, inappropriate bonding patterns involving these atoms, and the presence of unfavorable substructures such as three- or four-membered rings. We conduct 50 rounds of optimization and select the top 10 generated molecules from each round, resulting in 500 candidates. The summarized filtering rules and optimization conditions, along with the number of filtered generated catalysts, are reported in Supplementary Tables S11–S12. As shown in Fig. 7c, the optimized catalysts exhibit binding energies that mostly fall within the desired range. These optimization techniques effectively guide the generation process toward the desired target value.

To further validate these results, we select five Pd-complexes and perform DFT calculations using Gaussian16[60], following the parameter setup from the previous study[59]. The results showing the ligand composition and 3D structure of catalyst complexes are presented in Fig. 7d. The predicted binding energies closely match the DFT-calculated values. Note that, to ensure consistency with the previous study using the 3-21G basis set for DFT calculations, we also employed 3-21G for comparison, which resulted in good correlation. However, in the absence of computational time constraints, the use of more refined basis sets such as 6-31G(d,p) would be preferable. As a benchmark, calculations with the 6-31G(d,p) basis set, based on[61], were performed for five targets (Supplementary Table S13), revealing partial discrepancies compared to the 3-21G results. This suggests that incorporating high-precision basis sets into the evaluation of generated and filtered candidate catalysts would be valuable. When observing structural features, these catalyst complexes contain combinations of phosphine

ligands, which have been reported to play a crucial role in cross-coupling reactions[53,59]. Interestingly, many ligands contain fluorinated phosphines, which are notable structural features of Pd catalyst complexes and influence their electronic properties[62,63]. It should be noted that the model is trained based on a DFT-level dataset with structures where $L_1$ can be different from $L_2$ in Pd complexes. Although synthesis of those complexes remains challenging in practice, the results are encouraging and should be considered in future experiments. These findings support the potential of our model for catalyst design, validated by DFT calculation, highlighting its applicability to other catalytic systems.

**Asymmetric Pictet-Spengler (PS).** The final case study focuses on the asymmetric Pictet-Spengler (PS) reaction, which involves multiple conditions, including different species of catalysts and tryptamine derivatives with carbonyl compounds[46], as illustrated in Fig. 8a. This task relates to the study of asymmetric organocatalysts for enhancing enantioselectivity of catalysis, using experimental $\Delta\Delta G^{\ddagger}$ values curated from the literature as the target. The experimental $\Delta\Delta G^{\ddagger}$ values enable estimation of the enantioselectivity of catalyst-substrate combinations[46]. We construct a generative model integrated with surrogate models to predict $\Delta\Delta G^{\ddagger}$ values, and the model's performance on the dataset is shown in Fig. 8b.

To evaluate the model's usability in generating catalysts under different conditions, we randomly select a subset of conditions, which include combinations of two substrates, a solvent, and a co-catalyst, with more than 10 associated reactions. For each selected condition, the model generates ten thousand catalysts through random sampling. The filtering rules and the number of filtered generated catalysts for the displayed conditions are summarized in the Supplementary Tables S14–S15. We then compare the

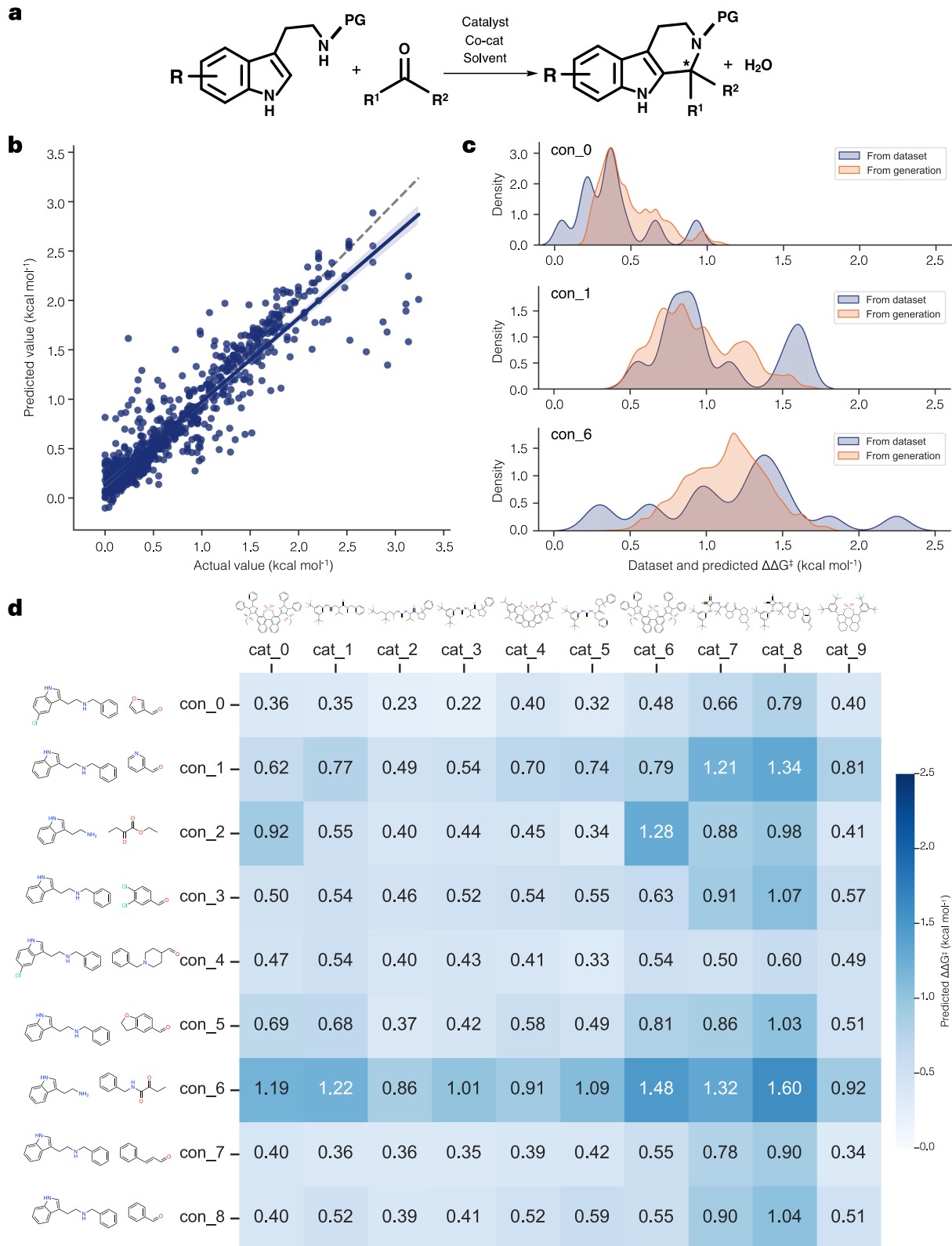

**Fig. 8 | Results on the case study of Asymmetric Pictet-Spengler (PS). a** Pictet-Spengler reaction of tryptamine derivatives and carbonyls[46]. PG is protecting group, H, or OH. **b** Predictive performance of the model on the entire dataset. The blue line represents the regression line, and the light blue area represents the 95% confidence interval for that regression. **c** Distribution of the dataset and predicted $\Delta\Delta G^{\ddagger}$ values from catalyst generation under three randomly selected conditions, visualized using kernel density estimation (KDE). **d** Heatmap showing predicted $\Delta\Delta G^{\ddagger}$ values for various generated novel catalyst-condition combinations. Generated catalyst structures are shown above the catalyst axis, while two substrates are only displayed along the condition axis to represent the reaction conditions.

distribution of predicted $\Delta\Delta G^{\ddagger}$ values with the corresponding dataset values for each condition, as shown in Fig. 8c. The results indicate that the model reasonably learns the target value distribution for each condition, and the predicted values of the generated catalysts align well with the dataset.

To further investigate, we identify common catalysts generated across different conditions. A heatmap of the predicted $\Delta\Delta G^{\ddagger}$ values for these common generated catalysts across conditions is shown in Fig. 8d. This heatmap apparently reveals variations in performance for each catalyst under different conditions. Notably, structures such as Cat_7 and Cat_8 exhibit promising performance for a wide range of reactions. These structures are similar to the Miller squaramide catalyst, which has been reported as an effective catalyst, particularly for reactions involving neutral indoles and aryl aldehydes[64]. This analysis highlights the model's capability to capture condition-dependent reactivity and demonstrates its potential in designing novel catalysts targeting generality with broad substrate scopes.

## Conclusions

In this study, we proposed a catalyst discovery framework called CatDRX, a reaction-conditioned generative model based on a variational autoencoder for catalyst generation and catalytic activity prediction. The model was pre-trained on a diverse reaction database and fine-tuned on downstream tasks. The model achieved promising performance in predictive tasks, particularly in yield prediction. Although some external catalytic activity datasets presented challenges due to their complex chemical spaces, the model demonstrated useful ability for extension to other endpoints. In terms of generative capability, the model could produce diverse catalysts with flexible options of sampling strategies. In addition, it showed favorable outcomes in real-world case studies, as validated by domain knowledge and computational chemistry tools, which can be possibly verified by the experiment. This highlights the potential of the model in facilitating catalyst discovery under particular reaction conditions.

A generative model often relies on large volumes of data, and this dependency could induce subtle challenges. The availability of high-quality and diverse chemical reactions was still limited, which affected the generalizability of various reactions and catalysts. Certain important reaction factors, such as temperature or concentration, and molecular descriptors, such as chirality for asymmetric catalysis, that were previously omitted, should be considered, as they may support specific tasks. In addition, this study faces limitations related to the lack of exploration of ligand conformational spaces, which can be crucial for understanding reaction transitions. Also, there are potential synthesizability issues, as some generated catalysts tend to exhibit complex structures or lack symmetry, making them more challenging to synthesize in practice.

For future work, several directions can be explored to improve the current framework. Integration of large ligand and catalyst complex libraries such as Kraken[65], OSCAR[66], ReaLigands[67], tmQMg-L[68], Clc-db[69], or OMol25[70] could positively enhance the model to capture structural diversity and molecular descriptors. A considerable portion of pre-training reaction data is excluded during acquisition due to the absence of well-defined catalyst roles. Thus, restoration of uncategorized reaction role[71] could help expand usable datasets. Regarding generation, refining the optimization approach can support generation, such as gradient-based methods and uncertainty quantification[72]. Apart from catalyst activity and selectivity, catalyst robustness, ability to maintain its structures and activity in extreme conditions, would be critical and need more consideration. Moreover, some catalysts have unique structures that pose significant synthetic challenges, such as ligands with asymmetrical phosphines[73]. Therefore, incorporating template- or modular-based generative strategies following recognizable backbones or scaffolds represents a favorable direction to elevate synthesizability[24,48,74].

In conclusion, this study presents a practical and effective methodology for generating catalysts tailored to specific chemical reactions. It can be extended to support future catalyst design, especially in the emerging trend of green catalysis, focusing on non-toxic and environmentally friendly materials. We believe that the proposed approach will contribute to the advancement of catalyst design and development, with particular relevance to industrial applications and the promotion of sustainable chemical manufacturing.

## Methods
### Reaction data

The scope of this study focuses on homogeneous catalysis, i.e., catalysts are in the same phase as the reactants and products. Homogeneous catalysts, particularly metal-ligand complexes, organometallic, or asymmetric catalysts, have been extensively utilized in various industrial processes due to their diverse reactivity and ability to control both activity and selectivity[75–77]. Additionally, this study is confined to single-step reactions yielding a single product, as reaction performance is evaluated using a single quantitative metric. Specific requirements for reaction data are established to guide data acquisition. Each reaction entry must include reactant(s), product, catalyst complexes, percent yield or another catalytic performance metric, and reaction time (in hours). In this context, both ligands and catalysts are considered part of the catalyst complex. Reagents, such as additives, solvents, and other auxiliary compounds, are optional. All compounds are represented using SMILES notation. Note that concentration, pressure, temperature, and other reaction conditions are not considered in this study due to challenges related to their availability and standardization across datasets. Although these conditions are essential for experiments, this study follows a common assumption that every reported reaction has been optimized for its conditions[36].

The datasets used in this study consist of a pre-training dataset and several downstream datasets. The pre-training dataset is sourced from the Open Reaction Database (ORD)[56], while the downstream datasets are collected from various sources. Essential pre-processing steps are applied to clean the reaction data. Full details of the datasets are presented in Supplementary Table S1.

*Pre-training dataset.* The pre-training dataset is from the ORD repository[56]. ORD provides a structured schema of information on organic reactions from various sources and categories, including high-throughput experiments and literature-extracted reactions. From approximately 2 million reactions, the data are processed using a simple user-friendly reaction format called SURF[78], which transforms the ORD schema files into a tabular format that facilitates standardization, promoting accessibility and usability. Next, reactions are selected based on the above requirements. Percent yield (%), measuring the ratio of the actual yield to the theoretical yield of the product in any type, is chosen as the performance metric. All SMILES notations are normalized, and reaction time must be greater than zero. For duplicate records containing the same reactants, products, catalysts, reagents, and reaction times, the average percent yield is calculated. Additionally, the molecular weight of the catalyst is included in the dataset. As a result, 57,823 unique reactions are extracted. Due to data imbalance and bias, outlier removal is performed using interquartile range (IQR) method based on catalyst molecular weight, the number of atoms in the catalyst, and percent yield. The percent yield values are then clipped to a range of 0 to 100[71]. To further reduce bias, overrepresented catalysts in the reaction data are downsampled. Finally, 52,448 reactions are selected for model pre-training, covering around 40,000 unique reactants and products and 2000 distinct catalysts. Although this amount of data set may be considered relatively small for pre-training, it still provides valuable diversity in chemical reaction knowledge for the model learning.

*Downstream dataset.* The downstream datasets are endpoint datasets measuring catalytic reaction performance, extracted from various sources of real-world chemical reactions. Downstream tasks include both percent yield and other catalytic activities related to reactivity and selectivity of catalysis, such as enantiomeric excess (ee, %), binding energy (kcal mol$^{-1}$), etc. These datasets are used in fine-tuning step to evaluate the model's prediction and generation performance.

### Data representation

Chemical reactions are represented with different formats depending on each type of component. All compounds involved in the reactions are

encoded using graph-based molecular representations, which effectively capture both molecular features and topological information. The proposed architecture utilizes two complementary views of graph-based representations.

For catalyst information, catalyst molecules are encoded in a matrix-based format, as this representation is both simple and convenient for reconstruction purpose. The matrix representation aligns naturally with graph connectivity and supports the decoding of multi-structured molecules, without being restricted to connected molecules as in tree-based graph approaches like JT-VAE[79]. The catalyst matrix contains three sub-matrices comprising a length column matrix, an annotation matrix, and an adjacency matrix, following the approach described in[80]. Formally, the matrix representation is defined as $\mathbf{M} = (T, A, B)$, where $T \in \{0, 1\}^S$ is the length column matrix, $A \in \{0, 1\}^{S \times A_t}$ is the annotation matrix, and $B \in \{0, 1\}^{S \times S \times B_t}$ is the adjacency matrix. Here, $S$ denotes the maximum number of nodes, $A_t$ is the number of atom types, and $B_t$ is the number of bond types. Catalyst molecules are inherently complex, often containing rare metal atoms and varying greatly in size and structure, necessitating adaptations beyond standard molecular configurations. In this study, the maximum number of nodes $S$ is set to 100. The atom types are extended to include 68 elements (including no-atom), and bond types are expanded to 6 (including no-bond). The length column matrix $T$ is encoded as a one-hot vector, indicating the size of the molecule for the reconstruction process. The annotation matrix $A$ is embedded as a one-hot vector to represent the atom type at each node, while the adjacency matrix $B$ uses a one-hot encoding to specify connections and bond types between nodes. As a result, the total dimensionality of the matrix representation is $100 + (100 \times 68) + (100 \times 100 \times 6) = 66900$ dimensions. The simplified illustration of catalyst matrix structure is displayed in Supplementary Fig. S1 and the features are summarized in Supplementary Table S2. Since reordering the nodes with bond rearrangement in a matrix representation can result in the same molecule, a data augmentation technique based on node order shuffling can be employed to increase the size of the dataset.

For other compounds in the reaction which are reactant, reagent, and product molecules are embedded as 2D graph representations, where atoms as nodes and bonds as edges. Each graph is defined as $\mathcal{G} = (\mathcal{V}, \mathcal{E})$, where $\mathcal{V} = \{v_0, v_1, \ldots, v_N\}$ is the set of $N$ nodes, and $\mathcal{E} = \{e_0, e_1, \ldots, e_M\}$ is the set of $M$ edges. Each edge is represented as a tuple $e = (v_i, v_j)$, where $i$ and $j$ are node indices indicating a bond between atoms. Node and edge features are extracted based on molecular properties as summarized in Supplementary Table S3, following the implementation of atomic graph from ref. [81].

Other conditions, such as catalyst molecular weights, reaction times, and catalytic activity targets, are represented numerically, with molecular weights as ranges and reaction times and catalytic activity targets as exact values.

## Model architecture

The model is designed based on joint-training or unified Conditional Variational Autoencoder (CVAE) integrated with property prediction, inspired by[82], with the utilization of reaction components as condition, as illustrated in Fig. 1b. It comprises three main modules: catalyst embedding module, condition embedding module, and autoencoder module including encoder, decoder, and predictor.

*Catalyst embedding module.* The catalyst matrix is embedded using the catalyst embedding module. The architecture of this part is displayed in Supplementary Fig. S2. Each component of the catalyst matrix is learned through a series of neural networks, and their outputs are subsequently summed to produce the final catalyst embedding. The resulting catalyst embedding is a vector $\mathbf{x} \in \mathbb{R}^{D_x}$, where $D_x$ denotes the size of the catalyst embedding.

*Condition embedding module.* Other compounds and reaction conditions are embedded using the condition embedding module. The molecules, including reactants, reagents, and product, are independently embedded using a Graph Attention Network (GAT). In cases where no reagent is present, a default embedding vector is used. Reaction time is represented

using its exact numerical value, while the molecular weight of the catalyst is encoded using a one-hot vector corresponding to predefined molecular weight clusters. The concatenated condition embeddings result in a vector $\mathbf{c} \in \mathbb{R}^{D_c}$, where $D_c$ denotes the size of the condition embedding.

*Autoencoder module.* The catalyst embedding and the condition embedding are concatenate to create the catalytic reaction embedding for autoencoder input. Autoencoder module contains encoder, decoder, and predictor.

- *Encoder:* The catalytic reaction embedding is learned through the encoder to produce a latent vector, denoted as $Q(\mathbf{z}|\mathbf{x}, \mathbf{c})$, where $\mathbf{z} \in \mathbb{R}^D$ and $D$ is the dimensionality of the latent space, following the general schema of CVAE.

- *Decoder:* A latent vector is sampled from the latent space and concatenated with the condition embedding to control the generation regarding to the reaction conditions. The latent vector with condition is then passed to the decoder module to decode into the catalyst matrix using all-atom generation technique. Thus, the decoder learns to reconstruct the catalyst molecule in matrix form, denoted as $P(\mathbf{M}|\mathbf{z}, \mathbf{c})$. The decoder module is divided into sequential steps with neural networks to facilitate the reconstruction as illustrated in Supplementary Fig. S3. First, the latent vector is passed to the length column matrix decoder, which contains a set of linear layers. Next, the latent vector, concatenated with the decoded length column matrix vector, serves as the input to the annotation matrix decoder. The annotation matrix decoder consists of multiple linear layers, with the final layer tasked with predicting the atom type for each node. Following this, the decoded annotation matrix is enhanced using another linear layer, and an outer product operation is performed to produce the initial adjacency matrix. The initial adjacency matrix is subsequently fed into a series of linear layers in the adjacency matrix decoder. The final layer of the adjacency matrix decoder predicts the bond type for each connection. Ultimately, the three decoded matrices are concatenated to reconstruct the final matrix representation of the catalyst. To support the decoding process, the concept of teacher forcing is integrated. During training, the true structures of the length column matrix, annotation matrix, and initial adjacency matrix are randomly used as starting points for decoding the next part at a fixed random ratio of 0.25. Finally, the decoded matrix representation are converted to molecular graph object using reconstruction process with post-processing.

- *Predictor:* Predictor $f(\mathbf{z}, \mathbf{c})$ accepts the mean latent vector of latent vector concatenated with condition embedding to predict catalytic reaction performance. During the fine-tuning step, the trained predictor within the framework is directly utilized to predict the percent yield. For other types of catalytic activity prediction, or when the original predictor performs poorly, an additional surrogate model is introduced during fine-tuning to handle the specific task. This surrogate model is different from the original predictor used during pre-training, but it also accepts the latent vector as input.

The loss function used in the training process is a combination of the standard VAE loss including the reconstruction loss and the Kullback-Leibler (KL) divergence loss and a regression loss for catalytic performance prediction. To balance the contribution of each component, additional weighting parameters, $\alpha$ and $\beta$, are introduced to assign appropriate importance to the reconstruction and KL losses, respectively. Totally, the loss is defined as $\mathcal{L}_{\text{total}} = \alpha(\mathcal{L}_{\text{recons}}) + \beta(\mathcal{L}_{\text{KL}}) + \mathcal{L}_{\text{prediction}}$.

*Reconstruction process.* The reconstruction process operates the conversion of the decoded catalyst matrix to a molecular graph. To enhance the validity of molecule generation, the post-processing step as a correction technique is implemented. This technique is adapted from similar works with all-atom reconstruction[83,84]. After reconstructing the atom types using the annotation matrix, bond types are determined to connect pairs of atoms. Instead of assigning bonds sequentially based on the order of atoms in the annotation matrix, the bond type probabilities from the adjacency matrix

are sorted in descending order and then assigned based on the highest probabilities. This approach prioritizes the model's confidence in connecting the most likely bonds between atom pairs. Next, after a bond is added to an atom pair, valency checking is performed to ensure the bond complies with chemical rules using RDKit functions. If a bond violates those rules, it is removed eventually. However, adding a charged atom is allowed in specific cases. Once the molecule is constructed, additional correction steps are applied, including post-valency checking, aromatic clearance, sanitization, and the removal of unconnected carbon atoms. These steps promote chemical validity of the final molecule. Note that although the generated molecule may differ from the originally decoded structure, post-processing corrections are designed to keep modifications minimal and chemically reasonable for aiding the generation of valid molecular structures.

## Training schema

In this study, a two-step training process is employed, consisting of a pre-training and a fine-tuning step. Firstly, the model is pre-trained using a pre-training dataset, which is divided into training, validation, and test sets with a ratio of 90: 5: 5. Data augmentation is applied to the training set with five shuffled node orderings. After pre-training, the model is fine-tuned for a specific task using downstream datasets. For the yield prediction dataset, fine-tuning is performed directly on the trained predictor. Meanwhile, for other catalytic activity predictions, an external surrogate model is introduced to accommodate different measurements. This approach enhances the model's transferability, allowing it to be extended to other catalytic activities across various types of catalytic systems. The hyperparameters are set separately for each dataset. The configuration parameters used during pre-training and hyperparameters for fine-tuning are provided in Supplementary Note 4. The model is evaluated using the results from the test sets. As all tasks are regression-based, the performance is assessed using root mean squared error (RMSE), mean absolute error (MAE), and coefficient of determination ($R^2$) metrics. We test all models using three different random seeds and the average values with standard deviation of performance metrics are reported.

## Catalyst generation

The generation of catalysts is performed after fine-tuning for a specific task. To generate a single catalyst, the latent space is sampled to obtain the latent mean, and the condition embedding is appended to generate the input for the decoder. Since our model incorporates reaction conditions resulting in a large latent space, we introduce three different sampling approaches to organize the search space effectively[82].

- *Random latent + Random condition:* The latent vector is randomly selected from the latent space, and the condition is randomly chosen from the training set. By default, the latent space will be constrained loosely to the latent space of the training set using integer bounding to control the quality, but it can be explored freely to the entire space.
- *Around sample + Random condition:* The latent vector is obtained by selecting a latent vector of a sampled catalyst and adding random noise. The condition is still randomly selected from the training set, which may not be related to the sampled catalyst.
- *Around sample + Sample condition:* The latent vector is obtained by selecting a latent vector of a sampled catalyst and adding random noise. The corresponding condition of that catalyst is also used as the condition embedding.

To assess the overall performance of catalyst generation, common molecule generation metrics are employed. Validity (Valid) is defined as the fraction of valid molecules based on the chemical validity-checking function. Uniqueness (Unique) is the fraction of valid and unique molecules within the generated set. Novelty (Novel) is the fraction of valid, unique, and novel molecules that are not present in the training set. In addition to general validity, task-specific validity (Valid (Task)) is introduced to evaluate the validity of generated molecules based on criteria specific to each task.

Internal diversity (IntDiv) measures the average pairwise molecular distance among valid and unique molecules, where a higher value indicates greater variety. Similarity to the nearest neighbor (SNN) evaluates the average molecular distance between each generated molecule and its closest counterpart in the training set, with higher values indicating stronger resemblance. Molecular distances here are calculated using the Tanimoto distance based on ECFP4 fingerprints (2048-bit). Lastly, the Fréchet ChemNet Distance (FCD)[85] is used to determine whether the generated molecules are both diverse and chemically or biologically similar to real molecules.

## Generation optimization

To generate the most promising catalyst under specific reaction conditions, an optimization process is employed. In this study, the Bayesian optimization algorithm from `scikit-optimize` (skopt)[86] is utilized to optimize the costly objective function associated with catalyst generation. The function to be evaluated involves several key steps. A latent vector is first sampled and concatenated with the given reaction condition embedding. This combined vector is then fed into the decoder to generate a catalyst molecule. The generated catalyst is subsequently embedded using the catalyst embedding module and re-encoded to obtain the corresponding latent representation. This catalyst latent vector, together with the condition vector, is input into the predictor to obtain the final predicted catalytic performance. The details of the parameters used in the optimization process are described in Supplementary Note 8.

The core objective of the optimization is to identify the catalyst that maximizes the predicted catalytic performance. Along with catalytic performance, other desirable or undesirable properties can be incorporated into the objective function. For example, reward terms can be added for satisfying certain favorable properties, while penalty terms can be applied if specific constraints are violated. This way provides a flexible framework to steer the generation process towards catalysts with more favorable characteristics.

In this study, two different search strategies are employed for optimization.

- *Search from random space:* The optimization process begins by sampling random latent vectors within the latent space, whose dimensionality is defined by the training set. This allows the optimizer to freely explore the chemical space and possibly generate a wide variety of candidate molecules. However, due to the huge chemical space, random exploration may struggle to identify optimal solutions efficiently.
- *Search around a starting molecule:* In this strategy, a specific target molecule is selected as the starting point. The search space is then constrained to a region around the latent vector corresponding to the target molecule. This approach enables controlled exploration within a localized neighborhood of the latent space.

## Generated catalyst validation using DFT

For the case study on the C-C cross-coupling (CC) dataset, we employ density functional theory (DFT) calculations to validate the catalyst complexes, following the methodology of the original study[59]. We use the transition-metal catalyst complex generation protocol from[87] to construct 3D coordinate structures by assembling the metal center, ligands, and substrate, together with calculating charge and multiplicity. DFT calculations are performed using Gaussian16[60], following the same protocol as the initial study[59]. In this work, we focus on Pd-complexes, thus geometry optimizations are carried out using the B3LYP-D3 functional with the 3-21G basis set, and single-point energies are computed at the B3LYP-D3/def2-TZVP level.

## Data availability

All datasets are publicly available in the original paper mentioned.

## Code availability

Codes are publicly available on GitHub http://github.com/ohuelab/CatDRX(https://doi.org/10.5281/zenodo.17164308).

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

## Acknowledgements

This work was financially supported by the Japan Science and Technology Agency FOREST (Grant No. JPMJFR216J), Japan Society for the Promotion of Science KAKENHI (Grant Nos. JP23H04880, JP23H04887, and JP23H04890), and Japan Agency for Medical Research and Development Basis for Supporting Innovative Drug Discovery and Life Science Research (Grant No. JP25ama121026).

## Author contributions

M.O. conceived the study. A.K. designed and implemented the methods and computed results. M.O. and A.K. analyzed the results. Y.K. and T.N. interpreted the results. A.K. drafted the manuscript and Y.K., T.N., and M.O. edited it. All the authors read and approved the manuscript.

## Competing interests

The authors declare no competing interests.
