## [Transparent Peer Review file · Communications Chemistry]

CatDRX as a reaction-conditioned generative model for catalyst design and optimization

Corresponding Author: Professor Masahito Ohue

Version 0:

Reviewer comments:

Reviewer #1

(Remarks to the Author)

Kengkanna and co-workers report a computational tool that will help advance the field of generative models for the discovery of homogeneous catalysts. The authors empathize the applicability of their framework (CatDRX) to multiple datasets involving different reaction components (reactants, products, catalyst classes), with a key aspect being a pre-training process on the Open Reaction Database. CatDRX appears to be well-designed and easily accessible, its implementation robust, and its predictive accuracy has been validated against other state-of-the-art model architectures. The manuscript is relatively well-written and easy to follow. However, certain sections would benefit from the use of more appropriate terminology to more strongly appeal to the homogenous catalysis community (e.g., be clear and precise about the nature of the distinct measures of "catalytic activity"). The manuscript's main limitation lies in sentences like "The results have been validated through computational chemistry and background knowledge from chemists" (p. 4). In practice, this translates to the computation of 5 binding energies (CC case study) and manually enumerated "synthetic accessibility/validity" rules. The impact of the manuscript would be much higher if experimental data on key catalyst/ligand candidates (e.g., optimized ligands reported in Fig. 4e) were collected. If this is beyond the authors' expertise or capabilities, I would highly recommend seeking the help of experimental collaborators.

Some additional comments are listed below:

1. The distinction between "predictor" and "surrogate predictor" is not always clear, nor is whether the "predictor" is always used (e.g., PS case study) or not, and for what purpose.
2. Figure 2: the nature of the target (e.g., yield, TOF, binding energy, $\Delta\Delta G^\ddagger$ etc...) for each dataset should be clarified.
3. Figures S6–S7 might be better suited for the manuscript (rather than the SI). Most readers might not be familiar with the content of ORD. A more detailed explanation of the type of reactions it contains is in order.
4. Figure 4 (and elsewhere): yields should always be reported as integer numbers.
5. Page 7: "according to the background in catalysis". As mentioned above, this represents one of the main limitations of the manuscript. Here (and elsewhere) is unclear what exactly the filtering principles are, how they are implemented, and how robust they are in defining what "good" catalyst candidates are.
6. In general, the ligand/catalyst structures depicted in Figures 4–6 are hard to see (specifically Fig. 6d). Higher quality renditions of the 2D structures should be used (e.g., ChemDraws). Overall, limited chemical insight into the structural features of the catalysts suggested by the algorithm is provided. Those sections should be expanded. Furthermore, it is difficult to discern whether catalysts/ligands generated by the algorithm are truly "novel" (e.g., the organocatalysts reported in Fig. 6d are included in the original dataset or not?)

Reviewer #2

(Remarks to the Author)

Summary

The authors report CatDRX, a catalyst discovery framework.

The overall architecture is centered around a conditional VAE, and is trained to perform two tasks: (1) encode/decode catalysts (represented as binary matrices) conditioned on contextual information about the corresponding reaction (molecular graphs + quantitative information), (2) predict downstream properties of the reaction via a predictor reading off the latent space of the VAE.

The generic (i.e. reaction-agnostic) architectural design enables the pre-training CatDRX on a moderately large dataset

(ORD). CatDRX can then be fine-tuned for reactions of choices, where the amount of available data is often more limited.

The performance and the design choices are evaluated on several datasets. Lastly, three case studies are presented to demonstrate the practical potential of CatDRX.

General comment

I am overall positive about the paper but further evidence could strengthen even further.

The paper tackles an important and timely problem: combatting the chronic issue of data scarcity in ML for QC and specifically in the context of generative modeling.

While CatDRX does not seem to outperform by a large margin more specialized baselines in predictive power, the proposed design is still relevant as it provides a potential blueprint for learning from different data sources while maintaining controllability at inference/sampling time through the proposed conditioning mechanism.

The paper is in general well-written and most of the claims are well supported by experiments. The experimental setup is rather extensive. However, while the predictive performance of the model is adequately studied via thorough ablations, I find that the generative capabilities of CatDRX could be better probed.

In particular, studying the impact of pre-training on the generative performance seems crucial to me, as this is one of the main selling points. An enlightening experiment would be to compare the generative performance of CatDRX when trained only on the downstream dataset vs. the pre-trained version. This would clearly show whether pre-training helps or not.

I would strongly recommend the authors to include such a study.

Questions

- In Section 2.4, how are the seed samples selected? -- this is important to specify that if the seed sample is taken from an area of the data that is not well represented in the training data, one might expect lower "performance".

- Latent representations of catalysts: As the encoder is not permutation invariant, two equivalent matrix representations of a given catalyst can lead to different latent representations. Did you find equivalent representations to cluster in the same parts of the latent space? Could that be an issue in particular when performing optimization in the latent space? Also, why is the encoder needed during optimization, could not the latent vector be optimized directly as the predictor operates directly in the latent space -- or did it lead to collapse / bad samples? Finally, the chosen value of beta is really small, implying that the learned latent space is probably far from Gaussian -- did you try other values of Beta or other ways to sample the latent space?

- Conditioning mechanism: What is the motivation for conditioning the encoder on the reaction context, if the condition is also provided directly as input to the decoder / predictor?

- Finetuning: Did you observe some form of forgetting during the finetuning phase -- i.e. that the benefits of the pre-training get lost when training for too long or with larger learning rates? Also, what part of the initial network is fine-tuned exactly? Section 2.1 says: "then fine-tuned on downstream datasets using a predictor or an additional surrogate mode" -- does that mean that only the encoder and predictor are fine-tuned, or also the decoder?

Minor comments

- What does CatDRX stand for -- as far as I can tell this is never explained?

- Fig S9a, I am not sure how it is constructed since the latent vector is drawn independently of the condition.

- In Section 2.5.1: "Although some deep learning models achieve the great results on the test set, they struggle to perform consistently on the training set." -- isn't the opposite based on Fig S12?

- Fig S10 could be moved to the main text, i.e. similar to Fig S2 for generation.

- A few typos:

* In Main, "reaction time" -> "reaction time";

* In Main, "boarder range" -> "broader";

* In Suppl., "valance" -> "valence".

- A few sentences would benefit from some extra polishing:

- * "Validation with background knowledge";
- * "to present the novelty"

Code and reproducibility

I did not run the code myself, but the repo is well organized, commented, and seems to be providing everything needed to reproduce the results presented in the manuscript.

Reviewer #3

(Remarks to the Author)

Kengkanna, Ohue and co-workers report a variational autoencoder for the inverse design of metal-based homogeneous catalysts. This is a hard problem, which is hereby tackled with a bold, well-thought approach achieving notorious results. Publication is recommended, but the following points should be considered by the authors for review:

- 1) For the $\Delta\Delta G^\ddagger$ prediction in the asymmetric catalysis case study: What happens if the catalyst and/or the substrate has more than one chiral center yielding different stereoisomers? It is stated that chiral centers are not encoded but not much is said about what are the consequences of this. Readers must know.
- 2) In section 2.4, besides validity, uniqueness, novelty, and diversity should be presented and discussed with more detail. It can't be all in the SI only. Also: What is the post-processing step? Explain it with more detail in the text, including the implications of it.
- 3) Regarding the subsection "Pre-training dataset": Why yields are clipped to 0-100? What does this mean? Shouldn't all yields be within this range by definition and otherwise be excluded as non-sense outliers? Clarify.
- 4) Section 4.7 addresses validation with DFT for the C-C cross-coupling - What about the other reactions? Also: Is it safe to do optimizations with the 3-21G basis set? Isn't it too small? Was it benchmarked against e.g. 6-31G? This is somehow unusual and needs justification.
- 5) In the Discussion, acknowledge that besides activity and selectivity, catalyst robustness (not addressed in this study) is also critical. Two other important things missing that are worth mentioning as potential limitations are: a) Exploration of the conformational spaces opened by the ligands, and b) Synthesizability, since generation tends to increase complexity and lack of symmetry.
- 6) Figure 2g: Why CatDRX is worse here? The justification given in the text is not very clear.
- 7) References: I missed a few references to the work of Kulik on catalysis (<https://doi.org/10.1021/jacsau.2c00176> and <https://doi.org/10.1021/acscatal.2c02096>) and Balcells, including the recent review (<https://doi.org/10.1021/acscatal.5c01202>) and the tmQMg-L ligand library (<https://www.nature.com/articles/s43588-024-00616-5>).

Congratulations to the authors: This is a sound approach to a very hard problem.

Version 1:

Reviewer comments:

Reviewer #1

(Remarks to the Author)

I believe the authors have addressed the majority of the issues I have raised in a way that is suitable for publication. I still believe some of the terminology they use is imprecise and the manuscript overall lacks chemical insight. For example, the sentences "When observing structural features, these catalyst complexes contain combinations of phosphine ligands, which have been reported to play a crucial role in cross-coupling reactions. Interestingly, many ligands contain fluorinated phosphines, which are notable structural features of Pd catalyst complexes and influence their electronic properties" is an incredibly vague answer that means very little. Although I believe CatDRX can be a useful tool, I would recommend thinking more deeply about the chemistry in future projects.

Reviewer #2

(Remarks to the Author)

I thank the authors for the revised article.

My concerns have been addressed and my questions answered.

Reviewer #3

(Remarks to the Author)

The manuscript was revised in depth and can now be published without any further changes.

Authors' Reply

To the review of

CatDRX: Reaction-Conditioned Generative Model for Catalyst Design and Optimization

Submitted to Communications Chemistry

Dear Editor and Reviewers,

We would like to express our sincere appreciation to the reviewers for their insightful and constructive feedback. The comments provided were instrumental in enhancing the clarity and quality of the manuscript. In response, we have undertaken a thorough revision and addressed each point raised. Presented below are the authors' detailed responses to the editor's and reviewers' comments.

Thank you for your time and kind consideration.

Yours sincerely,
Masahito Ohue

Replies to the editor's comments

(1) Transparency of filtering rules (Rev #1).

Provide the actual rule set or pseudo-code in the Supplementary Information, and add a short table showing how many candidates each rule removes in the three case studies.

Authors' reply:

Thank you for your recommendation. For the three case studies, we have provided the list of filtering rules used to select promising candidates for the generation process, as well as the conditions applied during the optimization process, in the Supplementary Information. In addition, we have included a summary table showing the number of remaining candidates after each filtering step, as your suggestion.

For Lewis acid-mediated Suzuki–Miyaura cross-coupling (L-SM) case study, the filtering rules and optimization conditions are displayed in Table S9. The summary table showing the number of candidates for each step of filtering are displayed in Table S10.

Table S9: Filtering rules and optimization conditions for L-SM dataset

Filtering rules	Optimization conditions
Number of fragment is 1 Contains P atom or N atom Number of neighbors of P is 3 P is not in 3-membered ring Number of neighbors of O does not exceed 2	Predicted yield is 100 Complies with the filtering rules Similarity to training set is not less than 0.25

Table S10: Number of filtered generated molecules from different generation approaches averaged from three different model seeds

Approach	From generation (around target molecule)	From generation (random in training space)	From optimization	
Total generated molecules	10000 (100.00%)	10000 (0.00%)	500 (100.00%)	
Total valid and unique molecules	5352 (53.52%)	8943 (89.43%)	342 (68.47%)	
Filtering rules	Number of fragment is 1	2716 (27.16%)	2170 (21.70%)	336 (67.20%)
	Contains P atom or N atom	2436 (24.36%)	1844 (18.44%)	336 (67.20%)
	Number of neighbors of P is 3	1251 (12.51%)	571 (5.71%)	331 (66.27%)
	P is not in 3-membered ring	1138 (11.38%)	512 (5.12%)	331 (66.27%)
	Number of neighbors of O does not exceed 2	1091 (10.91%)	466 (4.66%)	331 (66.27%)
Does not exist in training set	1077 (10.77%)	463 (4.63%)	321 (64.13%)	
Final candidate molecules	1077 (10.77%)	463 (4.63%)	321 (64.13%)	

For C–C cross-coupling (CC) case study, the filtering rules and optimization conditions for Pd-complex optimization are presented in Table S11. A summary table showing the number of candidates remaining at each filtering step is also provided as shown in Table S12. Since the filtering rules are applied during the optimization cycle to promote the generation of promising Pd-complex candidates, most of the generated complexes comply with these rules.

Table S11: Filtering rules and optimization conditions for CC dataset

Filtering rules	Optimization conditions
Number of fragments is 3 Contains metal atom of Pd Contains P, N, or O atom Number of neighbors of P is 3 Number of neighbors of N does not exceed 3 Number of neighbors of O does not exceed 2 Does not contain 3/4-membered ring	Binding energy at $-27.55 \text{ kcal mol}^{-1}$ Complies with the filtering rules Similarity to training set is not less than 0.2

Table S12: Number of filtered generated molecules for Pd-complex optimization

Approach		Optimization for Pd Complexes
Total generated molecules		500 (100.00%)
Total valid and unique molecules		422 (84.40%)
Filtering rules	Number of fragments is 3	282 (56.40%)
	Contains metal atom of Pd	282 (56.40%)
	Contains P, N, or O atom	282 (56.40%)
	Number of neighbors of P is 3	277 (55.40%)
	Number of neighbors of N does not exceed 3	276 (55.20%)
	Number of neighbors of O does not exceed 2	275 (55.00%)
	Does not contain 3/4-membered ring	268 (53.60%)
Does not exist in training set		245 (49.00%)
Final candidate molecules		245 (49.00%)

For Asymmetric Pictet–Spengler (PS) case study, the filtering rules are displayed in Table S14. There is no optimization process performed for this case study. The summary table showing the number of candidates for each step of filtering for specific reaction conditions are displayed in Table S15.

Table S14: Filtering rules for PS dataset

Filtering rules
Number of fragments is 1

Table S15: Number of filtered generated molecules for generation with specific reaction conditions

Approach		Generation based on condition		
		con_0	con_1	con_6
Total generated molecules		10000 (100.00%)	10000 (100.00%)	10000 (100.00%)
Total valid and unique molecules		6076 (60.76%)	6063 (60.63%)	7536 (75.36%)
Filtering rules	Number of fragments is 1	3290 (32.90%)	3409 (34.09%)	3880 (38.80%)
Does not exist in training set		3285 (32.85%)	3407 (34.07%)	3880 (38.80%)
Final candidate molecules		3285 (32.85%)	3407 (34.07%)	3880 (38.80%)

(2) Generative ablation results (Rev #2).

Extend the scratch-vs-pre-trained comparison to validity, uniqueness, novelty and diversity, similar to the request of R3. If space is limited, a single aggregate table in the SI is fine.

Authors' reply:

Thank you for your suggestion. We have extended the ablation studies on model generation performance by comparing a model fine-tuned on a pre-trained model with a model trained on the downstream dataset only, using the Asymmetric Pictet–Spengler (PS) dataset. We selected this dataset because it includes multiple catalysts and multiple reaction conditions. The results are summarized in Figure S10.

Overall, the validity of both methods is relatively high with the help of a post-processing module. In terms of uniqueness and novelty, the model fine-tuned on the pre-trained model produces a higher proportion of unique and novel catalysts than the model trained only on the downstream dataset, particularly when generating structures around known samples. This may be due to the broader catalyst knowledge learned from the pre-training set. For task-specific validity, the model trained only on the downstream dataset shows slightly higher values, likely because it was trained on a more limited scope. For diversity-related metrics, both models display similar trends. In summary, while the model trained on downstream dataset can achieve good validity in generated catalysts, it offers limited uniqueness and novelty compared to the fine-tuned model.

We have included this result in the manuscript and supplementary information.

Fig. S10: Model generation performance of ablation study comparing between model trained on pre- trained model and model trained on dataset only on Asymmetric Pictet–Spengler (PS) dataset with different sampling schema

(3) Quantitative evidence of (no) catastrophic forgetting (Rev #2).

Include a simple plot or table showing performance on a held-out pre-training test set over fine-tuning epochs.

Authors' reply:

Thank you for your suggestion. We have conducted the experiments again for the fine-tuning step, recording the learning performance on both the downstream test set and the held-out pre-training test set across each epoch. These experiments were performed on the SM dataset. Model performance was evaluated using L1-loss throughout the training process, as shown in Figure S7. Additionally, we provide the performance of the best-epoch model for your reference in Table S7. As observed, the model performs well on the fine-tuning downstream dataset but shows degraded performance on the held-out pre-training test set. This indicates signs of forgetting. As mentioned earlier, it would be an interesting future direction to explore fine-tuning strategies that balance performance on both the downstream and pre-training test sets to mitigate forgetting.

Fig. S7: Model learning performance over fine-tuning epoch measured in L1-loss comparing between downstream train, validation, and test set and held-out pretraining test set on SM dataset

Table S7: Model performance on held-out pre-training test set using different trained model stages

Model	Model performance		
	MAE	RMSE	R2
Model pre-trained on pre-training dataset	17.0572	23.09	0.379
Model fine-tuned on SM (Random) dataset	24.7436	30.6792	-0.0963
Model fine-tuned on SM (Test 1) dataset	24.2425	29.9675	-0.046
Model fine-tuned on SM (Test 2) dataset	24.2376	30.0517	-0.0519
Model fine-tuned on SM (Test 3) dataset	24.3269	30.1045	-0.0556
Model fine-tuned on SM (Test 4) dataset	24.1149	29.9005	-0.0413

(4) Chirality handling plan (Rev #3).

Authors explained that their representation does not allow for multiple stereocenters. It is unclear if the dataset contains such examples. In case yes, the authors should add a short paragraph quantifying how many reactions in the asymmetric catalysis set contain multiple stereocentres.

Authors' reply:

Thank you for your concerns. In the asymmetric hydrogenation (AH) dataset, all reactions (362) are annotated with either (R) or (S) chirality configuration in the product. Regarding the configuration of the catalyst, according to the original source of the dataset [28], the catalysts are consistently configured as (S) and the authors have accordingly considered an inverted product stereochemistry wherever experimental results were available only for the R-configured catalyst. Another related study [37] also notes that the ligands used in these reactions are axially chiral. However, all chiral ligands considered in this work share the same axial configuration; therefore, this aspect is not directly relevant in the present context.

For the asymmetric Pictet–Spengler (PS) dataset, the reactions were retrieved from multiple literature sources [46]. All catalysts are represented in a string-based format with a chiral carbon, but no explicit stereochemical configuration is indicated in the dataset. Regarding the dataset target, the goal is to predict enantioselectivity, which is measured by $\Delta\Delta G^\ddagger$. Therefore, the absolute configuration of the product is not directly relevant in this context.

We summarized this detail into short paragraph and added to the Result section (2.2 Catalytic Activity Prediction Performance), “In this study, for the AH dataset, according to the original sources [28,37], the catalysts are axially chiral but maintain an (S) chirality configuration and share the same axial configuration, making this aspect irrelevant here. For the PS dataset [46], all catalysts are represented in a string-based format with a chiral carbon but without explicit stereochemical configuration. The target property is enantioselectivity, measured by $\Delta\Delta G^\ddagger$, so the absolute product configuration is not directly relevant.”

(5) Basis-set benchmark (Rev #3).

Provide a small benchmark (e.g., ΔE for 5 Pd complexes) comparing 3-21G to a larger basis to be included in the SI.

Authors' reply:

Thank you for your information. We performed simulations using the same settings in Gaussian16 but with a larger basis set. According to the Gaussian16 documentation, the 6-31G basis set is only applicable to atoms from H to Kr and therefore cannot be used for palladium. Following the approach of [61], we used SDD for the Pd atom and 6-31G(d,p) for the other atoms. As shown, most results align well with previous experiments. Note that, since our model was trained using the original 3-21G setup, some differences may occur between the predicted values and those obtained with the larger basis set for certain complexes.

We added more explanation regarding this issue in the result section. "Note that, to ensure consistency with the previous study using the 3-21G basis set for DFT calculations, we also employed 3-21G for comparison, which resulted in good correlation. However, in the absence of computational time constraints, the use of more refined basis sets such as 6-31G(d,p) would be preferable. As a benchmark, calculations with the 6-31G(d,p) basis set, based on [61], were performed for five targets (Supplementary Table), revealing partial discrepancies compared to the 3-21G results. This suggests that incorporating high-precision basis sets into the evaluation of generated and filtered candidate catalysts would be valuable."

Table S13: Benchmark comparing binding energy (kcal mol^{-1}) between predicted values from CatDRX, calculated values from Gaussian16 with 3-21G basis set, and calculated values from Gaussian16 with SDD and 6-31(d,p) basis set

No	Complex	Predicted	DFT Calculated	
			Basis set: 3-21G	Basis set: SDD for Pd atom 6-31G(d,p) for others
1	[Pd].CP(F)F.Cc1ccc(P(c2ccc(C)cc2)c2ccc(C)cc2)cc1	-31.36	-31.74	-25.31
2	[Pd].CC(C)(C)CP(C(C)(C)C)C(C)(C)C.Clc1ccnnc1	-26.96	-26.31	-34.66
3	[Pd].Cc1cccc1P(c1cccc1C)c1ncccc1C.Fc1ccc(P(c2ccc(F)cc2)c2ccc(F)cc2)cc1	-26.29	-29.44	-25.20
4	[Pd].CP(C)c1cccnc1.FP(F)F	-25.97	-24.00	-23.45
5	[Pd].Cc1ccc(P(c2ccc(C)cc2)c2ccc(F)cc2)cc1.FCCNCCF	-28.99	-31.07	-41.51

Authors' reply:

We sincerely appreciate your encouragement and thoughtful feedback. We wish to thank the reviewers again for their valuable comments.

References:

- [28] Singh, S., Pareek, M., Changotra, A., Banerjee, S., Bhaskararao, B., Balamurugan, P., Sunoj, R.B.: A unified machine-learning protocol for asymmetric catalysis as a proof of concept demonstration using asymmetric hydrogenation. *Proceedings of the National Academy of Sciences* 117(3), 1339–1345 (2020)
- [37] Singh, S., Sunoj, R.B.: A transfer learning protocol for chemical catalysis using a recurrent neural network adapted from natural language processing. *Digital Discovery* 1(3), 303–312 (2022)
- [46] Gallarati, S., Gerwen, P., Laplaza, R., Brey, L., Makaveev, A., Corminboeuf, C.: A genetic optimization strategy with generality in asymmetric organocatalysis as a primary target. *Chemical science* 15(10), 3640–3660 (2024)
- [61] Ma, S., Cao, Y., Shi, Y. F., Shang, C., He, L., & Liu, Z. P. (2024). Data-driven discovery of active phosphine ligand space for cross-coupling reactions. *Chemical science*, 15(33), 13359-13368.

Replies to the reviewers' comments

Reviewer #1 (Remarks to the Author):

Kengkanna and co-workers report a computational tool that will help advance the field of generative models for the discovery of homogeneous catalysts. The authors empathize the applicability of their framework (CatDRX) to multiple datasets involving different reaction components (reactants, products, catalyst classes), with a key aspect being a pre-training process on the Open Reaction Database. CatDRX appears to be well-designed and easily accessible, its implementation robust, and its predictive accuracy has been validated against other state-of-the-art model architectures. The manuscript is relatively well-written and easy to follow. However, certain sections would benefit from the use of more appropriate terminology to more strongly appeal to the homogenous catalysis community (e.g., be clear and precise about the nature of the distinct measures of "catalytic activity").

Authors' reply:

We appreciate your constructive feedback. We have clarified some details about the reaction performance measurement in each case study. The revision can be found in the Result Section (Case studies subsection), highlighted in the revised manuscript.

The manuscript's main limitation lies in sentences like "The results have been validated through computational chemistry and background knowledge from chemists" (p. 4). In practice, this translates to the computation of 5 binding energies (CC case study) and manually enumerated "synthetic accessibility/validity" rules. The impact of the manuscript would be much higher if experimental data on key catalyst/ligand candidates (e.g., optimized ligands reported in Fig. 4e) were collected. If this is beyond the authors' expertise or capabilities, I would highly recommend seeking the help of experimental collaborators.

Authors' reply:

Thank you for your insightful comment. We agree that integrating experimental validation would significantly enhance the impact of our work. While our current study focuses on computational and expert-derived validation due to limitations in available time and resources, we recognize the importance of experimentally testing key catalyst/ligand candidates. As such, we consider this a valuable direction for future work. We are actively exploring opportunities to collaborate with experimental research groups to conduct synthesis and characterization of potential candidates identified by CatDRX. The collaboration would not only validate our optimization approach but also provide critical feedback to further improve the model's predictive capabilities and real-world applications.

Some additional comments are listed below:

1. The distinction between "predictor" and "surrogate predictor" is not always clear, nor is whether the "predictor" is always used (e.g., PS case study) or not, and for what purpose.

Authors' reply:

Thank you for your comment. The predictor is a prediction model trained during the pre-training stage specifically to predict reaction yields. When the model is fine-tuned for downstream tasks of yield prediction, the predictor is directly fine-tuned and used for making predictions. In contrast, the surrogate model is not involved during the pre-training stage. It is introduced during fine-tuning for downstream tasks involving other properties, or in certain cases of yield prediction when the original predictor performs poorly. The surrogate model also takes the latent vector from the VAE as input. For tasks involving prediction of other properties using the surrogate model, the predictions are not used for fine-tuning, as the targets differ from those in the original training. We have modified the description in the Method section (Model Architecture subsection) for clearer explanation.

2. Figure 2: the nature of the target (e.g., yield, TOF, binding energy, $\Delta\Delta G^\ddagger$ etc...) for each dataset should be clarified.

Authors' reply:

We appreciate the remark. We have updated the figure to include the target.

3. Figures S6–S7 might be better suited for the manuscript (rather than the SI). Most readers might not be familiar with the content of ORD. A more detailed explanation of the type of reactions it contains is in order.

Authors' reply:

Thank you for your suggestion. We have agreed to move the Fig S6-S7 to the manuscript and combine them into one figure as they represent a similar concept of chemical space.

We have added more explanation about ORD in the Method section (Reaction Dataset subsection). ORD provides a structured schema of information on organic reactions from various sources and categories, including high-throughput experiments and literature-extracted reactions.

4. Figure 4 (and elsewhere): yields should always be reported as integer numbers.

Authors' reply:

Thank you for pointing this out. We have modified yields in the figure to display as integer numbers

5. Page 7: "according to the background in catalysis". As mentioned above, this represents one of the main limitations of the manuscript. Here (and elsewhere) is unclear what exactly the filtering principles are, how they are implemented, and how robust they are in defining what "good" catalyst candidates are.

Authors' reply:

Thank you for your comments. The filtering rules are the fundamental criteria for assessing and selecting promising catalysts, based on basic chemical principles, synthetic accessibility scores, and desirable or undesirable structural features and properties. Currently, filtering is implemented independently for each task according to the nature of catalysts. It can be applied either after random generation or integrated into an optimization cycle. The implementation may consist of multiple if-

else conditions or score-based functions. In our case studies, filtering functions were designed based on chemists specializing in the relevant reactions and insights from the literature. These functions are flexible and can be customized for each task and user preference; as a result, the robustness of the filtering depends on the specific use case. Our approach identifies potential catalysts based on predicted catalytic activity, while allowing users to design appropriate filtering functions to select good candidates. We have revised the sentence as: "According to the background in catalysis and ligand availability, specific filtering principles are considered for identifying appropriate ligands for this reaction."

6. In general, the ligand/catalyst structures depicted in Figures 4–6 are hard to see (specifically Fig. 6d). Higher quality renditions of the 2D structures should be used (e.g., ChemDraws). Overall, limited chemical insight into the structural features of the catalysts suggested by the algorithm is provided. Those sections should be expanded. Furthermore, it is difficult to discern whether catalysts/ligands generated by the algorithm are truly "novel" (e.g., the organocatalysts reported in Fig. 6d are included in the original dataset or not?)

Authors' reply:

Thank you for your comments. Regarding the molecular images, we used ChemDraw to edit the 2D structures and generate vector-based images that can be scaled without any loss of quality.

About chemical insight into the structural features of the catalysts, we have expanded more detail about structure features in the CC dataset. "When observing structural features, these catalyst complexes contain combinations of phosphine ligands, which have been reported to play a crucial role in cross-coupling reactions [53,59]. Interestingly, many ligands contain fluorinated phosphines, which are notable structural features of Pd catalyst complexes and influence their electronic properties [62, 63]."

About the novelty of catalysts/ligands. The catalysts and ligands generated by the algorithm may include both novel structures and those already present in the training set. However, we ensure that all generated or optimized catalysts shown in the figures are novel and not included in the training data.

Reviewer #2 (Remarks to the Author):

Summary

The authors report CatDRX, a catalyst discovery framework.

The overall architecture is centered around a conditional VAE, and is trained to perform two tasks: (1) encode/decode catalysts (represented as binary matrices) conditioned on contextual information about the corresponding reaction (molecular graphs + quantitative information),

(2) predict downstream properties of the reaction via a predictor reading off the latent space of the VAE. The generic (i.e. reaction-agnostic) architectural design enables the pre-training CatDRX on a moderately large dataset (ORD). CatDRX can then be fine-tuned for reactions of choices, where the amount of available data is often more limited. The performance and the design choices are evaluated on several datasets. Lastly, three case studies are presented to demonstrate the practical potential of CatDRX.

General comment

I am overall positive about the paper but further evidence could strengthen even further.

The paper tackles an important and timely problem: combatting the chronic issue of data scarcity in ML for QC and specifically in the context of generative modeling. While CatDRX does not seem to outperform by a large margin more specialized baselines in predictive power, the proposed design is still relevant as it provides a potential blueprint for learning from different data sources while maintaining controllability at inference/sampling time through the proposed conditioning mechanism.

The paper is in general well-written and most of the claims are well supported by experiments. The experimental setup is rather extensive. However, while the predictive performance of the model is adequately studied via thorough ablations, I find that the generative capabilities of CatDRX could be better probed. In particular, studying the impact of pre-training on the generative performance seems crucial to me, as this is one of the main selling points. An enlightening experiment would be to compare the generative performance of CatDRX when trained only on the downstream dataset vs. the pre-trained version. This would clearly show whether pre-training helps or not.

I would strongly recommend the authors to include such a study.

Authors' reply:

Thank you for your interesting recommendations. In our ablation study, we previously compared the performance of a model fine-tuned from a pre-trained model (named Full CatDRX) with a model trained only on the downstream task (named NoAug+NoPT) using only the BH and SM datasets. We have now added additional ablation experiments on other datasets to more clearly demonstrate the performance differences between models fine-tuned from a pre-trained model and those trained

from scratch. The hyperparameter settings were derived from the fine-tuning with a pre-trained model. The results are shown in Fig. S6. The model fine-tuned from the pre-trained weights achieved better RMSE performance on many datasets, indicating that it benefits from the knowledge learned during the pre-training stage. These results have also been included in the Supplementary Information.

Fig. S6: Model prediction performance of ablation studies in RMSE comparing models trained using pre-trained model and models trained on dataset only

Questions

- In Section 2.4, how are the seed samples selected? -- this is important to specify that if the seed sample is taken from an area of the data that is not well represented in the training data, one might expect lower "performance".

Authors' reply:

Thank you for your concern. The three different seed samples were selected randomly and we added the word in the manuscript. We would like to evaluate the model under random situations without enforcing predefined training and testing sets. In many cases, reaction data is limited in size and coverage, often constrained to specific regions of catalyst space. Therefore, random splitting can somehow reflect this common situation. However, we also conducted out-of-sample splitting, where the types of reaction components are controlled between the training and testing sets, to assess the model's generalizability.

- Latent representations of catalysts: As the encoder is not permutation invariant, two equivalent matrix representations of a given catalyst can lead to different latent representations. Did you find equivalent representations to cluster in the same parts of the latent space? Could that be an issue in particular when performing optimization in the latent space? Also, why is the encoder needed during optimization, could not the latent vector be optimized directly as the predictor operates directly in the latent space -- or did it lead to collapse / bad samples? Finally, the chosen value of beta is really small, implying that the learned latent space is probably far from Gaussian -- did you try other values of Beta or other ways to sample the latent space?

Authors' reply:

Thank you for your question. To address the permutation invariance issue in equivalent matrix representations, we first represent each catalyst molecule as a normalized SMILES string before converting it to a matrix representation. This ensures that the encoder processes a consistent and unique representation for each molecule. During the optimization process, we also include the encoder to ensure accurate mapping of the generated molecules into the latent space, related to the following question. As a result, the predicted property values are derived from a single well-defined latent embedding. However, to further enhance permutation invariance, augmentation techniques and advanced equivariant learning methods may be required, which we leave as future work.

Secondly, the encoder is required during the optimization process because, after decoding, the molecular matrix undergoes a post-processing step to enhance chemical validity. This step may introduce slight modifications to ensure that the generated molecules comply with chemical rules. As a consequence, the final molecules may differ slightly from the initially decoded ones. To reflect these changes, we decided to encode the generated molecules with encoder again to obtain their actual latent vectors in order to acquire more accurate prediction of target properties.

Lastly, about the beta hyperparameter, we chose a lower value to encourage accurate reconstruction, given the diversity of catalyst structures, ranging from few atoms to larger multi-fragment systems. A smaller beta would help capture complex structural patterns and provide greater flexibility in the latent space. Moreover, since the model is jointly trained with a predictor module, this setup effectively balances both generative and predictive performance as well as exploration and exploitation. We experimented with larger beta values, but observed a decline in predictive performance. By the way, during the fine-tuning stage, the beta value was further adjusted depending on the specific task. We also applied beta-annealing to facilitate more stable and effective learning. These points have been incorporated into the supplementary information to improve clarity.

- Conditioning mechanism: What is the motivation for conditioning the encoder on the reaction context, if the condition is also provided directly as input to the decoder / predictor?

Authors' reply:

Thank you for your question. Our implementation is based on the original molecular generative model using a Conditional Variational Autoencoder (CVAE) [82]. The reaction condition is incorporated into both the encoder and decoder, enabling the embedding of condition alongside molecular structures. This guides the construction of the latent space by incorporating contextual information, which assists the generation process toward catalysts suited for specific reaction conditions.

In catalysis, reaction conditions are closely affected with catalytic efficiency, and catalysts often interact selectively with specific reactants. Our goal is to learn from a wide variety of reactions, and including reaction conditions in the encoder helps the model organize the latent space more meaningfully. This leads to the generation of more relevant catalysts, rather than producing random candidates that may be incompatible with certain reactions.

- Finetuning: Did you observe some form of forgetting during the finetuning phase -- i.e. that the benefits of the pre-training get lost when training for too long or with larger learning rates? Also, what part of the initial network is fine-tuned exactly? Section 2.1 says: "then fine-tuned on downstream datasets using a predictor or an additional surrogate mode" -- does that mean that only the encoder and predictor are fine-tuned, or also the decoder?

Authors' reply:

Thank you for your insightful point. To investigate the possibility of forgetting during the fine-tuning phase, we conducted an experiment comparing the prediction performance of a test dataset from the pre-training data using both the pre-trained model and the fine-tuned model on a downstream task. Our results showed a drop in performance of the fine-tuned model on the pre-training test dataset, which may indicate signs of forgetting during fine-tuning. We consider this effect as the focus of fine-tuning is to optimize performance on the downstream task and the structures of the catalyst may differ from pre-training. Nevertheless, we acknowledge the importance of this issue. It would be an interesting future direction to explore fine-tuning strategies that balance performance on both the downstream and pre-training test sets in order to mitigate forgetting.

The fine-tuning part of the initial network is the whole model including encoder, decoder, and also predictor as well. We have edited the description accordingly in the text, "then fine-tuned on downstream datasets using the whole pre-trained model including encoder, decoder, and predictor."

Minor comments

- What does CatDRX stand for -- as far as I can tell this is never explained?

Authors' reply:

Thank you for your question. CatDRX stands for Catalyst Discovery and RX, which refers to chemical reactions (ReaXions). We have edited the introduction to include this information, "CatDRX, Catalyst Discovery framework based on a ReaXion-conditioned variational autoencoder (VAE) for catalyst generation and catalytic performance prediction."

- Fig S9a, I am not sure how it is constructed since the latent vector is drawn independently of the condition.

Authors' reply:

Thank you for your question. The construction of the embedding space begins by randomly selecting ten conditions from the training set. For each condition, one thousand latent vectors are randomly sampled and used to generate catalysts through the generation process. These generated catalysts are then embedded with the corresponding condition to obtain latent vectors for visualization. Since the goal is to visualize the embedding space in a condition-dependent manner, we first convert the random latent vectors into catalysts, and then re-embed them with the specified condition. This ensures that the final latent vectors used in the visualization reflect both the catalyst structure and the associated reaction condition. We have updated the explanation for this part accordingly.

- In Section 2.5.1: "Although some deep learning models achieve the great results on the test set, they struggle to perform consistently on the training set. " -- isn't the opposite based on Fig S12?

Authors' reply:

Thank you for your remark. Yes, this sentence refers to Supplementary Fig. S11 and S12. We have now added references to these figures in the text.

- Fig S10 could be moved to the main text, i.e. similar to Fig S2 for generation.

Authors' reply:

Thank you for your suggestion. We have agreed to move Fig. S10, Model generation performance using different sampling schema on four datasets, to the main manuscript. However, we believe that Fig. S2, Catalyst embedding architecture, provides detailed information about the model architecture for catalyst embedding, which may not be directly related to the generation task. Therefore, we have decided to keep Fig. S2 in the Supplementary Information.

- A few typos:

* In Main, "reactiont time" -> "reaction time";

* In Main, "boarder range" -> "broader";

* In Suppl., "valance" -> "valence".

Authors' reply:

Thank you for your corrections. We have fixed all typos already.

- A few sentences would benefit from some extra polishing:

* "Validation with background knowledge";

* "to present the novelty"

Authors' reply:

Thank you for your comments. We have polished the phrases as follows:

Changed "Validation with background knowledge" to "validation based on reaction mechanisms and chemical knowledge"

Changed "to present the novelty" to "To promote novel catalyst generation"

Reviewer #3 (Remarks to the Author):

Kengkanna, Ohue and co-workers report a variational autoencoder for the inverse design of metal-based homogeneous catalysts. This is a hard problem, which is hereby tackled with a bold, well-thought approach achieving notorious results. Publication is recommended, but the following points should be considered by the authors for review:

1) For the $\Delta\Delta G^\ddagger$ prediction in the asymmetric catalysis case study: What happens if the catalyst and/or the substrate has more than one chiral center yielding different stereoisomers? It is stated that chiral centers are not encoded but not much is said about what are the consequences of this. Readers must know.

Authors' reply:

Thank you for raising this important point. Chirality is a critical factor in asymmetric catalysis. In our current study, due to limitations in embedding and featurization, chirality information was not explicitly included. As a result, the model may suffer from incomplete molecular representations, which can limit its predictive performance. More specifically, the lack of chirality encoding means that the model considers only one stereoisomer of a given catalyst–substrate pair. This focuses on a single product outcome, which may not fully reflect the stereochemical complexity of the system. If the analysis requires distinguishing between different stereoisomers, additional configuration or preprocessing would be necessary. One solution is to encode stereochemical information as part of the input conditions, similar to the approach [34], where stereoisomer assignment is incorporated into the graph-level features. We have added this explanation in the Results section (Catalytic Activity Prediction Performance subsection), “The current model does not include chirality information, so it can only focus on one stereoisomer of the outcome. Incorporating additional features could enrich representation and improve model learning. For example, encoding chirality configuration as part of the input conditions could enhance catalyst features, similar to the approach in [34].”

2) In section 2.4, besides validity, uniqueness, novelty, and diversity should be presented and discussed with more detail. It can't be all in the SI only. Also: What is the post-processing step? Explain it with more detail in the text, including the implications of it.

Authors' reply:

Thank you for your comment. We have discussed more details about uniqueness, novelty, and diversity in this section, “Regarding uniqueness and novelty, these two metrics exhibit similar trends and both reduce as a consequence of low validity. For diversity, although the trend is not clearly observed across all schemas, the generation process still produces high diversity, which is desirable for a generative model.”

The post-processing step is the modification and correction technique after molecule decoding to construct molecules with chemical rules to enhance validity. The detailed explanation about this step is described in the Method section, Model Architecture subsection, Reconstruction process paragraph. We have added a brief description and reference to this section in the manuscript, “The post-processing step involves

modification and correction techniques applied during molecule reconstruction using chemical rules to improve validity, as explained in detail in 4.3.”

3) Regarding the subsection "Pre-training dataset": Why yields are clipped to 0-100? What does this mean? Shouldn't all yields be within this range by definition and otherwise be excluded as non-sense outliers? Clarify.

Authors' reply:

Thank you for your comment. Clipping yields to the range of 0–100 means that any yield values below 0 are set to 0, and any values above 100 are set at 100. We acknowledge that, by definition, yield percentages should fall within this range. However, in practice, reported yields can occasionally exceed 100% due to experimental noise or impurities. To ensure broad coverage of reaction space during pre-training, we decided to retain these reactions instead of discarding them, while applying clipping to ensure numerical consistency, following the work from [71]. It is important to note that we performed outlier detection before this step using catalyst molecular weight, number of atoms in the catalyst, and percent yield. Therefore, reactions with yields over 100% represent only a small portion but useful in the cleaned subset of the data.

4) Section 4.7 addresses validation with DFT for the C-C cross-coupling - What about the other reactions? Also: Is it safe to do optimizations with the 3-21G basis set? Isn't it too small? Was it benchmarked against e.g. 6-31G? This is somehow unusual and needs justification.

Authors' reply:

Thank you for your question and concern. To perform validation with DFT, a comprehensive study of the full catalytic cycle, along with well-established benchmarks for comparison, is required. However, for the L-SM and PS datasets, the catalytic cycles are highly complex, and no reliable benchmarks are currently available to validate the calculations. Therefore, we limited DFT validation to datasets where benchmarks exist. However, we acknowledge that DFT-based simulation will be valuable tools for catalyst validation in the future.

Regarding the optimization of the CC dataset, our implementation follows the approach described in [49]. The original dataset construction employed this configuration to generate target values for Pd complexes. While the 3-21G basis set is relatively minimal, it has been shown to provide reasonable geometries and energies. To ensure consistency and allow for fair benchmark comparison, we retained the same configuration used in the original study.

5) In the Discussion, acknowledge that besides activity and selectivity, catalyst robustness (not addressed in this study) is also critical. Two other important things missing that are worth mentioning as potential limitations are: a) Exploration of the conformational spaces opened by the ligands, and b) Synthesizability, since generation tends to increase complexity and lack of symmetry.

Authors' reply:

Thank you for your insightful advice. We have incorporated your suggestions and revised the future work and limitations accordingly. These additions have been included in the Discussion section, “Apart from catalyst activity and selectivity, catalyst robustness, ability to maintain its structures and activity in extreme condition, would be critical and need more consideration.” and “In addition, this study faces limitations related to the lack of exploration of ligand conformational spaces, which can be crucial for understanding reaction transitions. Also, there are potential synthesizability issues, as some generated catalysts tend to exhibit complex structure or lack symmetry, making them more challenging to synthesize in practice.”

6) Figure 2g: Why CatDRX is worse here? The justification given in the text is not very clear.

Authors' reply:

Thank you for your question. Figure 2g shows the model performance on the C–C cross-coupling (CC) dataset. CatDRX appears to perform worse than other models. We have analyzed and identified several key reasons for this. First, regarding the reaction and catalyst space, as shown in Figures S6–S7, both the reactions and catalysts in the CC dataset locate outside the pre-training region. Additionally, the prediction target differs from that used during pre-training that is yield prediction, which limits the transferability of learned knowledge during fine-tuning. Second, in terms of reaction conditions and representation, the CC dataset includes only a single reaction condition. This causes the model to rely on catalyst structure alone and makes the model too complex. The matrix-based catalyst embedding, originally designed for generative tasks, may not be optimal for featurization in this context, limiting model performance. Alternative representations should be considered in future work. We have modified the justification to the following, “Similarly, in the catalyst space, a large portion of CC catalysts locate outside the pre-training region, further reducing the effectiveness of transfer learning. Moreover, the CC dataset contains only a single reaction condition, which limits the model’s ability to leverage condition-based knowledge and forces it to rely solely on the catalyst input. This results in an overly complex architecture and degraded performance.”

7) References: I missed a few references to the work of Kulik on catalysis (<https://doi.org/10.1021/jacsau.2c00176> and <https://doi.org/10.1021/acscatal.2c02096>) and Balcells, including the recent review (<https://doi.org/10.1021/acscatal.5c01202>) and the tmQMg-L ligand library (<https://www.nature.com/articles/s43588-024-00616-5>).

Authors' reply:

Thank you for your valuable recommendations. We have added suggested references in the Main and Discussion section.

Congratulations to the authors: This is a sound approach to a very hard problem.

Authors' reply:

We sincerely appreciate your encouragement and thoughtful feedback. We wish to thank the reviewers again for their valuable comments.

References

- [34] Aguilar-Bejarano, E., Özcan, E., Rit, R. K., Li, H., Lam, H. W., Moore, J. C., ... & Figueredo, G. (2025). Homogeneous catalyst graph neural network: A human-interpretable graph neural network tool for ligand optimization in asymmetric catalysis. *iScience*, 28(3).
- [49] Schilter, O., Vaucher, A., Schwaller, P., & Laino, T. (2023). Designing catalysts with deep generative models and computational data. A case study for Suzuki cross coupling reactions. *Digital Discovery*, 2(3), 728–735.
- [53] Cornet, F., Benediktsson, B., Hastrup, B., Schmidt, M. N., & Bhowmik, A. (2024). OM-Diff: inverse-design of organometallic catalysts with guided equivariant denoising diffusion. *Digital Discovery*, 3(9), 1793-1811.
- [59] Meyer, B., Sawatlon, B., Heinen, S., Von Lilienfeld, O. A., & Corminboeuf, C. (2018). Machine learning meets volcano plots: computational discovery of cross-coupling catalysts. *Chemical Science*, 9(35), 7069-7077.
- [62] Gioria, E., del Pozo, J., Lledós, A., & Espinet, P. (2021). Understanding the use of phosphine-(EWO) ligands in Negishi cross-coupling: experimental and density functional theory mechanistic study. *Organometallics*, 40(14), 2272-2282.
- [63] Sather, A. C., Lee, H. G., De La Rosa, V. Y., Yang, Y., Müller, P., & Buchwald, S. L. (2015). A fluorinated ligand enables room-temperature and regioselective Pd-catalyzed fluorination of aryl triflates and bromides. *Journal of the American Chemical Society*, 137(41), 13433-13438.
- [71] Sagawa, T., & Kojima, R. (2023). ReactionT5: a large-scale pre-trained model towards application of limited reaction data. *arXiv preprint arXiv:2311.06708*.
- [82] Lim, J., Ryu, S., Kim, J. W., & Kim, W. Y. (2018). Molecular generative model based on conditional variational autoencoder for de novo molecular design. *Journal of Cheminformatics*, 10(1), 31.